# Transcriptomic and Physiological Responses to Oxidative Stress in a *Chlamydomonas reinhardtii* Glutathione Peroxidase Mutant

**DOI:** 10.3390/genes11040463

**Published:** 2020-04-24

**Authors:** Xiaocui Ma, Baolong Zhang, Rongli Miao, Xuan Deng, You Duan, Yingyin Cheng, Wanting Zhang, Mijuan Shi, Kaiyao Huang, Xiao-Qin Xia

**Affiliations:** 1Institute of Hydrobiology, Chinese Academy of Sciences, Wuhan 430072, Hubei, China; maxiaocui_luck@126.com (X.M.); zhangbl@ihb.ac.cn (B.Z.); mronli@ihb.ac.cn (R.M.); dengxuan@ihb.ac.cn (X.D.); duanyou@outlook.com (Y.D.); cyy@ihb.ac.cn (Y.C.); zhangwanting@ihb.ac.cn (W.Z.); shimijuan@ihb.ac.cn (M.S.); 2Key Laboratory of Algal Biology, Institute of Hydrobiology, Chinese Academy of Sciences, Wuhan 430072, Hubei, China; 3University of Chinese Academy of Sciences, Beijing 100039, China

**Keywords:** glutathione peroxidase, oxidative stress, RNA-seq, reactive oxygen species, singlet oxygen, *Chlamydomonas reinhardtii*

## Abstract

Aerobic photosynthetic organisms such as algae produce reactive oxygen species (ROS) as by-products of metabolism. ROS damage biomolecules such as proteins and lipids in cells, but also act as signaling molecules. The mechanisms that maintain the metabolic balance in aerobic photosynthetic organisms and how the cells specifically respond to different levels of ROS are unclear. Glutathione peroxidase (GPX) enzymes detoxify hydrogen peroxide or organic hydroperoxides, and thus are important components of the antioxidant system. In this study, we employed a *Chlamydomonas reinhardtii* glutathione peroxidase knockout (*gpx5*) mutant to identify the genetic response to singlet oxygen (^1^O_2_) generated by the photosensitizer rose bengal (RB). To this end, we compared the transcriptomes of the parental strain CC4348 and the *gpx5* mutant sampled before, and 1 h after, the addition of RB. Functional annotation of differentially expressed genes showed that genes encoding proteins related to ROS detoxification, stress-response-related molecular chaperones, and ubiquitin–proteasome pathway genes were upregulated in CC4338. When GPX5 was mutated, higher oxidative stress specifically induced the TCA cycle and enhanced mitochondrial electron transport. Transcription of selenoproteins and flagellar-associated proteins was depressed in CC4348 and the *gpx5* mutant. In addition, we found iron homeostasis played an important role in maintaining redox homeostasis, and we uncovered the relationship between ^1^O_2_ stress and iron assimilation, as well as selenoproteins. Based on the observed expression profiles in response to different levels of oxidative stress, we propose a model for dose-dependent responses to different ROS levels in *Chlamydomonas*.

## 1. Introduction

Plants and algae are the main producers of atmospheric oxygen, which inevitably leads to the production of a large amount of reactive oxygen species (ROS) within cells. Photosystem releases oxygen with H_2_O, accompanied by the production of ROS, which include singlet oxygen, superoxide, hydrogen peroxide, and hydroxyl radicals [1,2,3]. ROS play a dual role in plant physiological processes, potentially causing damage (membrane lipid peroxidation, DNA damage, and irreversible hyperoxidation of proteins) but also functioning as indispensable short- and long-range signaling molecules in numerous cellular responses [4,5]. In vascular plants, ROS are involved in the regulation of development, cell differentiation, intracellular redox levels, signal transduction, stress responses, and induction of apoptosis [4,6,7,8,9,10]. Therefore, ROS homeostasis must be carefully controlled to support signaling while limiting damage to cells. 

To protect themselves from damage caused by oxidative stress and to redirect the actions of ROS into signaling pathways, plants have evolved a diverse set of antioxidant factors, including enzymes that detoxify ROS. For example, the non-selenocysteine glutathione peroxidase GPX5 detoxifies hydrogen peroxide with cytosolic thioredoxin TRX-*h*1, not glutathione (GSH), as the electron donor [11,12]. In *Chlamydomonas reinhardtii*, singlet oxygen (^1^O_2_) produced from the photosensitizers neutral red, methylene blue, or rose bengal (RB) induces a remarkable upregulation of *GPX5* expression. Hydrogen peroxide or the superoxide-generating compound paraquat induces a somewhat weaker upregulation of *GPX5* [11,13,14]. A *C. reinhardtii gpx5* knockout mutant showed depressed synthesis of lipid droplets compared with its parental strain CC4348 [15], suggesting that a high level of ROS attenuated lipid-droplet formation. 

Research in algae and vascular plants has investigated the signaling pathways that modulate gene expression in response to ^1^O_2_ stress, including direct and retrograde (i.e., chloroplast-to-nucleus) signaling. For example, the cytosolic zinc finger protein METHYLENE BLUE SENSITIVITY (MBS) is conserved in *Arabidopsis* and *Chlamydomonas* and affects the oxidative stress response [16]. Screening for *Chlamydomonas* lines that lack the acclimation response to ^1^O_2_ identified the *singlet oxygen acclimation knocked-out 1* (*sak1*) mutant [17]. SAK1 is a putative transcription factor and is hyperphosphorylated during ^1^O_2_ acclimation. How these signaling pathways act together to modulate the response to ^1^O_2_ stress remains unclear.

In this study, we employed the unicellular green alga *C. reinhardtii* to investigate ^1^O_2_ signaling and responses to oxidative stress in the *gpx5* mutant. To this end, we sequenced the transcriptomes of the parental *C. reinhardtii* strain CC4348, the *gpx5* mutant, and the complemented strain L27 before and after treatment with RB. By comparing the transcriptomes, we elucidated how ^1^O_2_ signaling regulates gene transcription and the role of the key antioxidant enzyme, GPX5, in this response. 

## 2. Materials and Methods 

### 2.1. Strains and Culture Conditions

The parental strain of *C. reinhardtii* was CC4348, which was also available as *sta6* (*cw15 nit1 NIT2 arg7-7 sta6-1:ARG7 mt^+^*) strain descripted as from the *Chlamydomonas* Resource Center (http://www.chlamycollection.org). The *sta6* mutant has been shown not only defective for the small subunit of the heterotetrameric ADP-glucose pyrophosphorylase [18,19], being also affected in respiratory burst oxidase gene (RBO1, Cre03.g188300) [20] and sensitive to oxidative stress levels of lipid peroxidation [21]. Because of high oil accumulation under N-starvation, CC4348 was initially used to screen the mutants with defects in lipid droplets formation, the *gpx5* mutant was one of them. The *gpx5* mutant and its complemented strain L27 were obtained as described previously [15]. As the initial concentration at 1.7 × 10^6^ cells·mL^−1^ (OD_750_ = 0.5), all cells were cultured mixotrophically in Tris-acetate-phosphate (TAP) medium (20 mM Tris, 0.4 mM MgSO_4_∙7H_2_O, 0.34 mM CaCl_2_∙2H_2_O, 10 mM NH_4_Cl, 1 mM phosphate, 10 mL/L glacial acetic acid and trace elements [22]) at 25 °C under constant incident illumination of 60 µmol·m^−2^∙s^−1^ with continuous shaking at 120 rpm. Cells were grown on solidified TAP medium containing 1.5% agar at 25 °C under constant incident illumination of 60 µmol·m^−2^·s^−1^ for about 5 days. 

### 2.2. RB treatment Assays and Sampling

For the RB treatments, 100 mL of liquid culture was harvested at a density of 2 × 10^6^ cells mL^−1^ (about OD_750_ = 0.6) by centrifugation at 2500× *g* for 3 min at room temperature. The supernatant was discarded, the cells were washed in fresh TAP medium one time, and then they were resuspended in an equal volume of fresh culture medium and grown at 25 °C under constant incident illumination of 60 µmol· m^−2^·s^−1^ with continuous shaking at 120 rpm, and fresh RB (Sigma Aldrich, Missouri, USA) was added to a final concentration of 1 µM. Samples taken immediately upon inoculation with RB-containing media were labeled as 0 min, and subsequent samples were taken after 15, 30, 45, and 60 min incubation and labeled accordingly. Only samples at 0, 30, and 60 min were used for RNA-seq, while all were used for validation by quantitative real-time PCR. This experiment was performed in triplicate. 

### 2.3. ROS Measurement

Cells were stained with 2′,7′-dichlorofluorescein diacetate (DCFH-DA) (Sigma Aldrich, Missouri, USA) as described in our previous study [15]. The nonpolar, nonionic DCFH-DA becomes nonfluorescent DCFH after crossing cell membranes and being enzymatically hydrolyzed by intracellular esterases. Hydrogen peroxide oxidizes DCFH to fluorescent dichlorofluorescein (DCF). The fluorescence of DCF was detected by using a multi-mode microplate reader (Filter Max F5, Molecular Devices; excitation at 485 nm and emission at 530 nm). The ROS abundance after treatment with RB was quantified by the following equation: (F_TAP+**RB**_)/F_TAP_. Under oxidative stress condition, the level of ROS was normalized for each strain from untreated condition independently.

### 2.4. Isolation of Total RNA

Ten mL sample cultures (at a density of 2 × 10^6^ cells ml^−1^) were collected by centrifugation for 3 min at 2500× *g* and 4 °C, washed in RNase-free water, and resuspended in 5 mL of Trizol Reagent (Invitrogen, CA, USA). The suspension was rocked for 5 min before it was flash-frozen in liquid nitrogen and stored at −80 °C. After all samples were collected, the samples resuspended in Trizol were taken out, thawed at 24 °C for about 10 min. In proportion to the volume of Trizol, 1 mL of chloroform was added to the suspension, rocked for 15 s, following by maintaining on ice for 3 min. After centrifuged at 12,000× *g* for 15 min at 4 °C, the supernatant was saved to add 2.5 mL of isopropanol, followed by maintaining on ice for 10 min. After centrifuged at 12,000× *g* for 10 min at 4 °C, the supernatant was washed by 75% ethanol, dried, and dissolved in diethyl pyrocarbonate-treated double-distilled H_2_O (ddH_2_O). After removal of the final traces of ethanol, the pellet was resuspended in 50 µL of H_2_O containing 0.1% diethyl pyrocarbonate. The RNA concentration was measured on a NanoDrop 2000 spectrophotometer (Thermo Fisher Scientific, MA, USA), and the quality of the RNA was assessed on an Agilent 2100 Bioanalyzer (Agilent Technologies, CA, USA).

To analyze the gene expression of the whole-genome, RNA samples from CC4348, the *gpx5* mutant, and the complemented strain L27 were isolated and used to construct the sequencing libraries for Illumina sequencing. Briefly, mRNA was purified from 5 μg of total RNA using oligo (dT) magnetic beads, broken into 450–550 bp fragments, and added unique adapter for every sample, using the TruSeq RNA sample preparation kit (Illumina, CA, USE) according to the manufacturer’s instructions. 

### 2.5. RNA-Sequencing, Mapping, and Gene Expression Analyses

For estimating the transcript abundance in the RB incubation time-course experiments, the pooled sequencing libraries were sequenced on the Illumina HiSeq X-ten platform (Illumina Inc., CA, USA) for 150 bp pair end sequencing mode. About 10 G of sequencing data were obtained per library for each sample. After filtering low-quality reads with a Perl script (IlluQC.pl) from the NGSQC Toolkit [23], high-quality reads were mapped to the *C. reinhardtii* version 5.5 genome (Department of Energy JGI), using TopHat software [24]. No more than two mismatches per sequencing read were allowed. More than 10 million reads with unique mapping were mapped for each sample, and the uniquely mapped ratio was about 90% of total mapped reads in each sample. Unique mapping reads were assigned to *C. reinhardtii* version 5.5 transcripts, to record the mapped count number, using HT-Seq [25]. Analyses of differential expression, including FDR calculations, were performed by using the Bioconductor package edgeR [26]. Expression estimates were in units of RPKM [27] normalized by library size and mappable transcript length. If the RPKM of one transcript was equal to zero, the minimum value in that experiment was used to replace it. The sets of genes that met the following three conditions in each comparison were selected for further analysis: (1) RPKM log_2_ ratios were considered significant if ≥2 or ≤−2; (2) at least one RPKM ≥ 10; (3) the *p-*value for differential expression was set to ≤0.05.

### 2.6. Quantitative Real-Time PCR

RNA was isolated from samples after treatment with 1 µM RB immediately (0 min) and after 15, 30, 45, and 60 min. The cDNA was transcribed by using First-Strand cDNA transcription kit (Thermo Fisher Scientific, MA, USA) according to the manufacturer’s protocol. The cDNA was then diluted 1:3. Quantitative real-time PCR (qRT-PCR) was performed, using specific primers (Appendix A) and LightCycler 480 SYBR Green (Applied Biosystems, CA, USA) according to manufacturer’s protocol on the Applied Biosystems 7500 Fast real-time PCR system. The sizes of amplification products were 100 to 300 bp. All samples were run in duplicate, with CT value normalized to *CβLP*. The 2^−∆∆CT^ method [28] was employed to calculate relative fold differences. 

### 2.7. Measurements of Photosynthetic Efficiency 

Photosynthetic efficiency was measured as the PSII maximum quantum yield, using the Fv/Fm parameter. Then, 3 mL of culture, at a density of 1 × 10^6^ cells/mL, was collected and dark-adapted for 15 min. Fv/Fm was measured by using a multiple excitation wavelength modulated chlorophyll fluorometer (Heinz Walz Gmbh Effeltrich, Wurzburg, Germany).

### 2.8. Measurements of Chlorophyll Content

For chlorophyll content, including chlorophyll *a* and chlorophyll *b*, 4 × 10^6^ cells were collected into a 2 mL centrifuge tube and centrifuged at 12,500 g for 1 min. The precipitate was dissolved in 1 mL of methanol for 24 h in darkness, at 4 °C. After centrifugation at 12,500× *g* for 2 min, 750 μL of supernatant was used to measure absorbance values at 665.2 and 652.0 nm, using a spectrophotometer. The chlorophyll content was determined according to the following equation [29]: chlorophyll content (μg/mL) = 2.71 × OD_665.2_ + 22.12 × OD_652.0_.

### 2.9. Measurements of Percentage of Cells with Flagella and Flagella Length

CC4348 and its derivative *gpx5* mutant lack a cell wall and are difficult to be crossed to obtain the cell with flagella, so widely used wild-type CC125 strain was employed for measuring flagella length. After treatment with 1 μM RB for 30 min, aliquots of CC125 cells were fixed with 0.5% Lugol’s solution. Images were recorded with a microscope (Nikon ECLIPSE Ti-U, Tokyo, Japan) equipped with a camera (Digital Sight DS-U3, Nikon, Tokyo, Japan) with an LWD 40× objective lens. At least 100 flagella were measured, and 200 cells were counted, and the ratio of the ciliated cells was calculated, using the NIS-Elements BR (version 4.10) software. This experiment was performed in triplicate.

## 3. Results

### 3.1. GPX5 Maintains ROS Homeostasis in C. reinhardtii

Expression of *C. reinhardtii GPX5* is strongly induced by the ^1^O_2_ generated from the photosensitizer RB [11,13,14]. To investigate the molecular mechanism by which GPX5 helps maintain ROS homeostasis, we treated the parental strain CC4348, the *gpx5* knockout mutant, and the complemented strains L27 or C203 [15] with 2 μM RB for 1 h. Then, the total intracellular ROS content was measured, using the fluorescent dye 2′,7′-dichlorofluorescein diacetate (DCFH-DA). Compared to the untreated cells, in CC4348 the level of ROS increased and reached a maximum level (2.1-fold) after 1 h treatment with RB and then gradually decreased to the initial level after 18 h. In the *gpx5* mutant, the level of ROS reached a maximum level (3.3-fold) after 1 h treatment and then gradually decreased to the initial level after 18 h. The maximum ROS level in the *gpx5* mutant was significantly higher than that in CC4348. In the complemented strain L27, the level of ROS increased to 2.5-fold, which was intermediate between the parental strain and the mutant (Figure 1A). These results demonstrated that GPX5 plays an important role in the removal of ROS. 

To explore the effect of extra ROS on growth, cells of CC4348, the *gpx5* mutant, and two complemented strains (L27 and C203) were treated with 2 or 4 μM RB for 1 h. After treatment, the cells were spotted on solid TAP medium, and their growth was monitored after 4 days. As expected, there was no difference in growth among the four strains without RB treatment (Figure 1B). By contrast, for cells that had been treated with 2 or 4 μM RB, the spots of the CC4348 strain were much larger and greener after 4 days, compared with the *gpx5* mutant. Indeed, the *gpx5* mutant grew more slowly after being treated with 2 μM RB compared to the parental strain and was unable to survive 4 μM RB treatment. The two complemented strains, L27 and C203, grew similarly to CC4348. These data indicated that GPX5 plays an important role in maintaining the growth of *C. reinhardtii* cells recovering from oxidative stress. 

In order to quantify the effect of GPX5 on growth without RB treatment, we measured the growth rate of CC4348, *gpx5* mutant, and L27 cells cultured in TAP complete medium under optimized laboratory growth conditions. CC4348 cells reached a stable period after 72 h of incubation, and the OD_750_ reached 1.6. However, the growth of *gpx5* cells was compromised, and the OD_750_ was only 1.2 when it reached a stable stage. The growth status of L27 was in between that of CC4348 and *gpx5*, and the OD_750_ was 1.4 when it reached a stable stage (Figure 1C). We also found the good correlation between the cell number and OD_750_ as shown in Appendix A. Lower OD_750_ in the stable period of *gpx5* mutant suggested the lower cell density. These data indicated that the deficiency of GPX5 effected cell cycle and lowered the population density.

### 3.2. The Transcriptome of gpx5 Mutants Under Laboratory Growth Conditions

To assess the transcriptome of the *gpx5* mutant under optimized laboratory growth conditions (no extra oxidative stress added), we performed RNA-seq on CC4348 and *gpx5* cells grown at a density of 2 × 10^6^ cells mL^−1^ in exponential phase in TAP complete medium at 25 °C under constant illumination of 60 µmol·m^−2^∙s^−1^ with continuous shaking at 120 rpm. The experimental design and analyses are shown in Appendix A. The sequence reads were aligned, and expression level were determined based on the *Chlamydomonas* reference genome [30,31] (Appendix A). Differentially expressed genes (DEGs) were screened with the criteria of false discovery rate (FDR) ≤ 0.05, fold change ≥ 2, and at least half of transcript abundance from pairwise comparisons ≥ 10 reads per kb of exon per million fragments (RPKM). According to these criteria, we identified 880 DEGs in the *gpx5* mutant compared to the parental strain CC4348 under optimized growth conditions, of which 380 DEGs were upregulated and 500 were downregulated (Appendix A).

These DEGs were classified by using MapMan [32], employing the Algal Functional Annotation Tool [33] (Figure 2A). We first analyzed transcripts involved in redox metabolism, among which the abundance of 8 transcripts increased, and the abundance of 14 transcripts decreased in the *gpx5* mutant compared to CC4348. The *GPX5* transcript was hardly detectable, as expected. *GPX4*, another member of the GPX family, had a slightly higher transcript abundance in the *gpx5* mutant than in CC4348. The other members of GPX family, *GPX1–GPX3*, had no significant change in transcript abundance in the *gpx5* mutant. These data suggested that the expression of other GPX proteins were not upregulated to compensate for the lack of GPX5.

Among the eight upregulated transcripts related to redox in the *gpx5* mutant under optimized growth conditions, the transcripts of two nucleoredoxins (*NRX2* and *NRX3*) and one sulfiredoxin (*SRX*1) increased by more than 4.5-fold. As antioxidants, NRXs reduce other proteins by cysteine thiol-disulfide exchange, functioning as protein disulfide reductases (PDIs). Similarly, *PDI5* transcript abundance significantly increased more than two-fold in the *gpx5* mutant. NRX2, NRX3, SRX1, and PDI5 can reduce the disulfide bond of oxidized proteins, attenuating the effects of oxidative stress. The abundance of transcripts of *VTC2* (Cre13.g588150), encoding GDP-L-galactose phosphorylase, a key enzyme in de novo ascorbate synthesis [34], increased by about three-fold in the *gpx5* mutant, suggesting that the synthesis of the antioxidant ascorbate (ASC) was induced. These data suggested that some mechanisms of ROS scavenging were enhanced in the *gpx5* mutant, and that, even under unstressed mixotrophic conditions, the presence of ROS occurred and was sufficient for cell signaling.

In order to detect the effects of GPX5 deficiency on basal metabolism, we found the modifications in the CCM occurred in the mutant. Most of genes encoding proteins involving in CCM were upregulated in the *gpx5* mutant, including some low-CO_2_-inducible proteins (CCP2, LCI24, LCI5, LCIB, and LCIE). Moreover, we also analyzed the expression change of genes in the chlororespiraton. Among them, seven out of nine transcripts showed no difference in abundance between CC4348 and the *gpx5* mutant, including glycolate dehydrogenase (GYD1, Cre06.g288700), phosphoglycolate phosphatase (PGP1 and PGP2 encoding by Cre03.g168700 and Cre10.g438100, respectively), serine hydroxymethyltransferase (SHMT1, Cre16.g664550), alanine-glyoxylate transaminase (AGT1 and AGT2), and serine glyoxylate aminotransferase (SGA1). Only hydroxypyruvate reductase (HPR1) and alanine aminotransferase (AAT1) were downregulated in the *gpx5* mutant. These results meant the deficiency of GPX5 had few effects on chlororespiraton.

Transcript abundances of the glutathione *S*-transferase (GST) genes (*GST1* and *GST10*) were very low in the *gpx5* mutant (4 and 6 RPKM, respectively) compared to that of either GST1 or GST10 in CC4348. GSTs detoxify endogenous or xenobiotic compounds and prevent oxidative damage to cells [35,36,37]. Stress-inducible GSTs with glutathione peroxidase activity can protect plants from oxidative injury [38]. Moreover, GSTS are biomarkers of the response to stress and overexpression of GSTs enhances tolerance to abiotic stress [39,40]. The transcript abundance of *GSTs* in the *gpx5* mutant was not higher than that in CC4348. This suggests that the *gpx5* mutant was not experiencing oxidative stress when grown on TAP complete medium under our optimized laboratory growth conditions.

Mutation of *GPX5* resulted in the adjustment of metabolic processes. The transcripts of 38 photosynthesis-related genes were decreased and 14 photosynthesis-related transcripts were increased in the *gpx5* mutant. Most of the transcripts encoding antenna proteins showed a downward trend, except the *LHCA3* transcript, encoding a chlorophyll a/b binding protein of light-harvesting complex (LHC) I, which increased about 2-fold in abundance in the *gpx5* mutant. Moreover, nearly all of the genes involved in tetrapyrrole synthesis were downregulated, except for *UPS1*, which encodes uroporphyrinogen-III synthase (Cre09.g409100) (Figure 2A,B). To confirm the effect of GPX5 deficiency on photosynthesis, we measured the maximum efficiency of photosystem II (PSII) (Figure 2D). The results showed a slightly reduction in photosynthetic efficiency of the *gpx5* mutant compared to CC4348, suggesting that the *gpx5* mutant might minimize ROS accumulation by slowing down photosynthesis, which is a major source of ROS. 

Among 37 DEGs encoding proteins related to carbon metabolism, including glycolysis, gluconeogenesis, starch synthesis, starch degradation, and the TCA cycle, 30 transcripts were downregulated in the *gpx5* mutant. The transcript with the largest decrease in level encodes malate synthase (MAS1), which uses acetyl coenzyme A to synthesize malic acid. The abundance of the *MAS1* transcript decreased from 756 RPKM in CC4348 to 221 RPKM in the *gpx5* mutant. We also detected the isocitrate lyase (ICL1) transcript, which decreased in abundance from 785RPKM in CC4348 to 282 RPKM in the *gpx5* mutant under laboratory growth conditions; and so did the transcripts of citrate synthase (CIS2) and acetyl-CoA syntheses/ligase (ACS3). The transcript of CIS2 and ACS3 decreased from 80 to 62 RPKM, and from 2508 to 1361 RPKM, respectively. These data would simply suggest that, since *gpx5* is growing more slowly than the control, the transcripts encoding transcripts involved in the glyoxylate cycle are in lower amounts. Almost all genes encoding enzymes in the glycolysis/gluconeogenesis pathway were downregulated in the *gpx5* mutant, except fructose-1,6-bisphosphate aldolase (FBA1). Nevertheless, the reaction from fructose-1,6-diphosphate to glyceraldehyde-3-phosphate catalyzed by FBA1 is reversible, so it might not affect the overall metabolic slowdown of the glycolysis/gluconeogenesis pathway. 

Lack of GPX5 also affected lipid metabolism, resulting in 27 lipid-related DEGs (9 upregulated and 18 downregulated). The transcript abundance of *KAS2* and *KAS3*, which encode β ketoacyl acyl-carrier-protein synthase II and synthase III, respectively, and *HAD1*, encoding 3-hydroxyacyl-carrier-protein dehydratase, decreased to 0.07- to 0.35-fold in the *gpx5* mutant. The transcripts of *FAD6*, encoding chloroplast omega-6-fatty acid desaturase, increased in abundance by 2.3-fold, reaching 1374 RPKM. Increasing the unsaturation of membrane lipids in cyanobacteria can improve the tolerance to low temperature [41]. Therefore, upregulation of the expression of genes encoding fatty acid desaturases may affect the metabolism of unsaturated fatty acid in the *gpx5* mutant.

Functional classification showed that 15 out of 18 DEGs related to amino acid metabolism were downregulated in the *gpx5* mutant. The abundance of transcripts encoding the enzymes involved in arginine synthesis were significantly decreased, including arginosuccinate synthase (Cre09.g416050, AGS1), acetylglutamate kinase (Cre01.g015000, AGK1), argininosuccinate lyase (Cre01.g021251, ARG7), N-acetylglutamate synthase (Cre16.g694850, NGS1), and N-acetyl-gamma-glutamyl-phosphate reductase (Cre03.g146187). These observations suggested that decreasing synthesis of arginine might play some function in response to GPX5 deficiency or the resulting ROS imbalance.

Besides the effect of oxidative stress on metabolism, transcripts related to the cell cycle and structural proteins were significantly downregulated in the *gpx5* mutant (Figure 2A,C). These effect on cell-cycle-related genes is consistent with the effect on growth described above (Figure 1C). In our previous study [15], we found that the *gpx5* mutation died earlier than the parental stain CC4348 when cultured in nitrogen-free medium. The grow difference between the mutant *gpx5* and parental strain suggested that the deficiency of GPX5 in the mutant not only affected ROS homeostasis, but also affected cell growth.

We also identified DEGs related to nucleotide or nucleic acid metabolism, transcript regulatory, mitochondrial electron transport chain (mETC) or encoding transporters. For example, the transcripts encoding a bZIP transcription factor BLZ8 (Cre12.g501600) and a zinc finger transcription factor (Cre03.g156250) increased about 2.5-fold, with transcript abundance increasing from 108 and 105 RPKM in CC4348 to 258 and 253 RPKM in the *gpx5* mutant, respectively. 

In summary, intracellular antioxidant metabolism and transport appeared to adjust to maintain redox homeostasis in the *gpx5* mutant. The expression of genes related to carbon metabolism, lipid metabolism, and synthesis of chlorophyll were depressed, and the efficiency of photosynthesis decreased in the *gpx5* mutant rather than in CC4348. These overall multifaceted changes in transcription slowed the growth of the *gpx5* mutant cells.

### 3.3. Transcriptome Responses to ^1^O_2_ in Parental Strain CC4348 Cells 

To address the role of GPX5 in response to oxidative stress, we first studied the mechanisms by which parental strain *C. reinhardtii* cells respond to oxidative stress. To this end, we performed RNA-seq on CC4348 cells at 0, 30, and 60 min after the addition of 1 μM RB to the medium. Using the same criteria as before, we identified 708 DEGs after the 30-min RB treatment, accounting for 4.7% of the total genome. These DEGs included 322 upregulated transcripts and 386 downregulated transcripts (Appendix A).

ROS homeostasis was disturbed in CC4348 cells under oxidative stress. The transcripts of 22 redox metabolism-related genes were upregulated, and six were downregulated (Figure 3A). The transcript abundances of genes encoding ROS-scavenging enzymes, such as ascorbate peroxidase (APX1), glutathione reductase (GSR1), GPX5, and the GST family, were upregulated after ^1^O_2_ stress. *GPX5* has been reported to be highly induced by ^1^O_2_ [11,12], and the transcript abundance of *GPX5* increased 13.5-fold after treatment with RB for 30 min in CC4348 cells (Appendix A). Another non-selenocysteine GPX gene (*GPX4*) and selenium-dependent GPX gene (*GPX2*) showed no difference in transcript abundance. Additionally, the transcript of *GPX3* was barely detectable. Interestingly, one selenium-dependent GPX gene (*GPX1*) showed the opposite expression pattern, as it was downregulated from 100 to 6 RPKM after 30 min of RB treatment. The active site of GPX1 is selenocysteine, which is easily oxidized and has strong reducibility. Generally, antioxidants function actively during oxidative stress, so the downregulation of *GPX1* was unexpected. One possible explanation is that different *GPX* genes are induced by different oxidative stresses.

We noted that most genes involved in redox metabolism were upregulated in CC4348 cells after treatment with 1 μM RB for 30 min. Among the 28 redox-related DEGs, the transcripts of *ferritin2*, encoding a ferritin-like protein (Cre01.g033300) had the greatest fold change, from 14 to 1944 RPKM, an increase of about 140-fold (Figure 3B and Appendix A). Ranked second were SOUL heme-binding protein genes (*SOUL1* and *SOUL2*), whose transcripts increased more than 60-fold. This indicated that heme might play an important role in maintaining redox homeostasis. Moreover, *GST* family genes also responded to RB treatment, including Cre07.g342100, *GST1* (Cre16.g688550), *GST10* (Cre12.g559800), and *PDO2* (Cre12.g559800) which belong to the GST KAPPA class. Among them, Cre07.g342100 encodes a GST, whose transcript increased sharply by 29-fold within 30 min after ^1^O_2_ stress, reaching a peak of 121 RPKM. Thioredoxin superfamily proteins and proteins containing thioredoxin domains, including NRX2, protein disulfide-isomerase A6 (PDIA6, Cre07.g326600), and r53.5-related protein LCI7 (Cre06.g263550) also responded to ^1^O_2_. Among them, *PDIA6* and *LCI7* were upregulated more than 15-fold within 30 min of the treatment. 

To confirm the RNA-seq data, we performed qRT-PCR on a set of DEGs, including the *GPXs*, *APX1*, *Mn superoxide dismutase* (*MSD2*), *SOUL1*, *SOUL2*, *LCI7*, and the *GSTs* (Figure 4). The expression pattern from qRT-PCR was similar to that from RNA-seq, and the coefficient of association between them was about 0.84 (Appendix A). Both RNA-seq data and qRT-PCR results showed that most of the transcript abundances of enzymes to detoxify ROS were upregulated. Overall, after the RB treatment, the antioxidant system was activated in CC4348 cells via the expression of related genes, providing sufficient antioxidant proteins to eliminate the ^1^O_2_ stress generated by RB.

RNA-seq data also show that treatment with RB induced the expression of an iron-assimilating protein gene (*FEA1*, Cre12.g546550) and stress-related molecular chaperones, which assist in proper protein folding. The transcript abundance of *FEA1* increased to 1703 RPKM after 30 min of stress from an initial value of 590 RPKM. Molecular chaperones also responded strongly. For example, the transcript abundance of *HSP70A* increased about 3-fold, from 261 to 726 RPKM, while HSP90A increased more than 3-fold, from 275 to 901 RPKM after treatment with RB for 30 min, respectively (Figure 3C and Appendix A). Genes encoding heat shock proteins (HSP22E, HSP22F, HSP70C, HSP70E, HSP90B, and HSP90C), CLPB3 (a ClpB chaperone belong to HSP100 family, Cre02.g090850), HOP1 (HSP70-HSP90 organizing protein), CPN60C (Chaperonin 60C), a DnaJ-like protein DNJ34, and HSP90-cochaperone (Cre07.g341550) were also all upregulated in CC4348 cells after ^1^O_2_ stress for 30 min. This upregulation of the expression of molecular chaperones could be due to the accumulation of incorrectly folded or oxidatively damaged proteins due to oxidative stress. In support of this conclusion, we found that the transcript abundance of 17 proteasome-related genes, encode proteins involved in ubiquitination-dependent proteasome-degradation pathways, including ubiquitin-activated enzyme E1 (UBA1), ubiquitin ligase E3 (Cre12.g533750, Cre12.g500550, Cre12.g501450, and Cre11.g476250), and seven proteasome subunits, increased more than 2-fold. In addition, the transcripts of genes encoding seven proteases or metalloproteinases were upregulated. 

We found 27 DEGs related to carbon metabolism, including glycolysis, the TCA cycle, and oxidative pentose phosphate pathway (OPP), and expression of 25 genes was upregulated after treatment with RB for 30 min. Four rate-limiting enzymes, a fructose-bisphosphatase (FBP1), phosphofructokinases (PFK1 and PFK2), and a pyruvate kinase (PYK4) involved in glycolysis were all induced. Notably, the *PYK4* transcript increased about 5-fold to peak at 112 RPKM. Transcripts encoding another three key enzymes involved in the TCA cycle, 2-oxoglutarate dehydrogenase, citrate synthase, and isocitrate dehydrogenase, were also upregulated more than 2.5-fold. Only carbonic anhydrase (CAH8) and chitinase (Cre07.g317250) showed a different expression pattern relative to other proteins involved in carbon metabolism. On the whole, glycolysis and the TCA cycle pathway showed a rising trend, which might provide more ATP to respond to oxidative stress. 

The expression of 21 transcripts related to lipid metabolism was upregulated, and only two genes were downregulated. Transcripts encoding cyclopropane fatty acid synthase (*CFA1* and *CFA2*) increased in abundance by about 19-fold and 32-fold, to 217 and 70 RPKM, respectively (Figure 3B,C, Appendix A). The transcript of a sterol-sensing 5-transmembrane protein (*SSD1*) could hardly be detected under normal growth conditions, but after 30 min of RB treatment, the *SSD1* transcript reached 10 RPKM. SSD1 can function as a lipid transporter; therefore, the increase of *SSD1* transcript implied that lipid composition changed in response to oxidative stress.

Enriched classes of DEGs also included transcripts related to hormone metabolism. For example, seven enzymes involved in jasmonic acid (JA) biosynthesis were induced after RB treatment for 30 min, including Acyl-CoA oxidase (*ACO2*, *ACO3*, and *ACO4*) and 3-hydroxyacyl-CoA dehydrogenase (*HCD1*), acyl-coenzyme A thioesterase 9 (*TEH6*, Cre16.g683350), lnoleate 13S-lipoxygenase (*13-LOX*, Cre12.g512300). Notably, *OPR* encoding 12-oxophytodienoic acid reductase (Cre03.g210513) increased in transcript abundance by about 6-fold, from 38 to 226 RPKM. The identity between 13S-lipoxygenase (13-LOX, Cre12.g512300) in *C. reinhardtii* to lipoxygenase (LOX1, AT1G55020) in *Arabidopsis thaliana*, 12-oxophytodienoic acid reductase (OPR, Cre03.g210513) in *C. reinhardtii* to oxophytodienoate-reductase 3 (OPR3, AT2G06050) in *Arabidopsis thaliana* is 36% and 46%, respectively, so we are convinced to conclude that jasmonic acid biosynthesis occurred in *C. reinhardtii*. JA regulates the induced defense responses to heavy metals in the green alga *Chlorella vulgaris* [42] and cold tolerance in *Arabidopsis* leaves [43]. Therefore, we speculated that JA might also play an important role in responding to RB treatment in *C. reinhardtii*.

In response to RB treatment, 94% of the genes encoding cell-structure-related proteins were downregulated. Moreover, 88% (134 DEGs) of the downregulated genes encoded flagellar associated proteins (FAPs), including Intraflagellar Transport (IFT) proteins. To further explore whether oxidative stress affected the expression of FAPs, we examined CC125 cells treated with 1 μM RB for 1 h. Because CC4348 strain is the mutant without flagella, widely used CC125 strain was used to measure the length and ratio of flagella. Up to 37% of these cells lost their flagella at 12 h. The length of flagella did not change (Appendix A), suggesting that RB induced deflagellation, not resorption.

Overall, ^1^O_2_ activated the detoxication of ROS, stimulated increases in the abundance of HSP family proteins and ubiquitin-mediated proteolysis, upregulated carbon metabolism and lipid metabolism, and induced deflagellation. 

### 3.4. Transcriptome Responses to ^1^O_2_ in the gpx5 Mutant

To address the function of GPX5 in response to oxidative stress, we compared the transcriptomes of the parental strain CC4348 and the *gpx5* mutant during ^1^O_2_ stress. RNA-seq was performed on the parental strain CC4348, the *gpx5* mutant, and complemented strain L27 at 0, 30, and 60 min after the addition of 1 μM RB (Figure 5). Using the same screening criteria, we identified 1852 DEGs in CC4348 or the *gpx5* mutant under oxidative stress. To compare the responses of CC4348 and the *gpx5* mutant, we divided the 1852 DEGs into three categories. Group A were upregulated during the oxidative stress in CC4348 and included 947 genes. Group B did not have significantly different expression during the treatment and had 478 genes. Group C were downregulated during the response to ^1^O_2_ in CC4348. An overview of the functions related to these transcript categories is presented in Table 1, with an emphasis on differences between CC4348 and *gpx5* mutant cells (Appendix A).

The group A DEGs were further classified into three subgroups, A1 (362 genes, upregulated in CC4348 and *gpx5*), A2 (50 genes, upregulated in CC4348, not changed in *gpx5*), and A3 (15 genes, upregulated in CC4348 and downregulated in *gpx5*) (Figure 5A). The highest enriched categories of the subgroup A1 proteins include ROS detoxification, carbon metabolism, and ubiquitin-dependent proteolysis. In subgroup A1, 31 transcripts encoded proteins regulating redox homeostasis, including APX1, GSR1, GST6, GST10, VTC2, TRXs, and SRX1. These antioxidases or antioxidant proteins responded to oxidative stress independent of GPX5.

Although the transcript abundance of most genes related to carbon metabolism was lower in the *gpx5* mutant than that in CC4348 under normal growth conditions, transcripts encoding proteins related to glycolysis and the oxidative pentose phosphate pathway (OPP), such as FBP1, PFK1, PFK2, and transaldolases (TAL1, TAL2) from subgroup A1, were upregulated in *gpx5* in the same pattern as in CC4348 after RB treatment. In particular, the *TAL1* transcript increased from 26 to 194 RPKM in *gpx5* and from 89 to 250 RPKM in CC4348. Increasing glycolysis metabolism and OPP would likely produce more NADPH to counteract the oxidative environment in cells. 

Genes from subgroups A2 and A3 (upregulated in CC4348 and unchanged or downregulated in *gpx5*) are mainly involved in chlorophyll metabolism or related to redox or transport. Eight transcripts encoding proteins involved in chlorophyll synthesis from subgroup A2 were significantly increased in abundance in CC4348, whereas those mostly remained constant in the *gpx5* mutant after RB treatment. Magnesium chelatase subunit transcripts (*CHLD*, *CHLI1*, and *CHLI2*) were upregulated by more than 2-fold (peak more than 50 RPKM) in CC4348 after treatment, but did not significantly change in the *gpx5* mutant. Subgroup A3 had only one transcript related to chlorophyll synthesis, *CPX1* (coproporphyrinogen III oxidase), which was upregulated in CC4348 but downregulated in the *gpx5* mutant. These observations suggested that the chlorophyll content would increase in CC4348 but not in *gpx5* after RB treatment.

Five transcripts from subgroup A2 and 3 transcripts from subgroup A3 encoded putative transporters, including transporters in vesicular trafficking (Cre04.g224800 and Cre09.g394954), mitochondrial oxoglutarate/malate carrier protein (Cre11.g467535), sodium-exporting ATPase (Cre17.g744447), and major facilitator superfamily transporter (MFT10, Cre02.g095076) from subgroup A2, and sodium/bile acid cotransporter 7 (Cre02.g095086), MFT26 (Cre12.g512200), and a solute carrier family 35 member (Cre06.g286200) from A3. 

We also observed that the abundance of transcripts encoding thioredoxin peroxidase (Cre16.g654250) from subgroup A2 and NRX2 from subgroup A3 increased in CC4348 under oxidative stress, but not in the *gpx5* mutant. The transcript for a 60-kDa SS-A/Ro ribonucleoprotein (Cre17.g725750) from subgroup A3 increased in abundance, from 35 to 93 RPKM in CC4348 after RB treatment for 30 min, but decreased from 136 to 37 RPKM in the *gpx5* under the same conditions. This RNA-binding protein is differentially expressed depending on GPX5 under oxidative stress, suggesting that it regulates the downstream response. 

Genes in group B were similarly classified into subgroups, according to their response in the *gpx5* mutant under oxidative stress. Among them, 382 genes, which were designated as B1, were upregulated in the *gpx5* mutant, and 565 genes, designated B3, were downregulated in the *gpx5* mutant (Figure 5B). Therefore, genes in subgroup B1 were specifically upregulated in the *gpx5* mutant, and genes in subgroup B3 were specifically downregulated in the *gpx5* mutant.

Unlike the parental strain CC4348, the *gpx5* cells had high concentrations of ROS after RB treatment (Figure 1A). The high concentration of ROS induced transcripts from subgroup B1, which are mainly associated with the mETC, carbon metabolism, protein metabolism, and endosomal sorting complex required for transport (ESCRT). The transcript abundance of genes encoding cytochrome c oxidase subunits (*COX2A* and *COX3*), cytochrome c oxidase assembly factor (*SCO1*), ubiquinone cytochrome c oxidoreductase subunit (*QCR1*), and eight mitochondrial ATP synthase subunits in the mETC remained stable in CC4348, but increased by 2- to 5-fold in the *gpx5* mutant after oxidative stress treatment. Besides, we found a type II NAD(P)H dehydrogenase (NDA1), an alternative oxidase (AOX1), complex IV (SCO1 and PET191) upregulated, and this fits well the data published by Sabeeha S. Merchant [44]. They showed transcripts encoding proteins of the most mETC genes not changed and the O_2_ consumption decreased. These results suggested that ATP produced by respiration is in smaller amounts in stress conditions. 

Transcripts encoding proteins involved in carbon metabolism, especially the TCA cycle and starch metabolism, were also enriched in subgroup B1. Indeed, 12 transcripts encoding proteins involved in the TCA, phosphoenolpyruvate carboxykinase (*PCK1*), mitochondrial pyruvate dehydrogenase complex alpha subunit (*PDC1*), isocitrate dehydrogenase (*IDH1* and *IDH2*), Dihydrolipoamide succinyltransferase (*OGD2*), succinyl-CoA ligase (*SCL1* and *SCL2*), succinate dehydrogenase (*SDH1* and *SDH3*), fumarase (*FUM1*), and NAD-dependent malate dehydrogenase (*MDH3* and *MDH4*) were significantly increased only in the *gpx5* mutant after RB treatment for 30 min. In addition, *OGD1*, *CIS1*, and *IDH3* increased in the *gpx5* mutant and CC4348. The upregulation of the TCA cycle, along with increased mETC activity, likely supply more energy to respond to oxidative stress in the *gpx5* mutant.

A comparison of *gpx5* and CC4348 also showed that transcripts related to ubiquitin-mediated proteolysis were induced after oxidative stress both in CC4348 and *gpx5* cells, including 30 transcripts from subgroup A1 and 33 transcripts from subgroup B1. Almost all transcripts related to the ubiquitin–proteasome system increased in abundance during the 60-min oxidative stress treatment in CC4348 (Appendix A). Under the control conditions (0 min), the abundance of these transcripts was lower in the *gpx5* mutant than that in CC4348. After 60 min of RB treatment, these transcripts showed a dramatic upregulation in the *gpx5* mutant, significantly larger than that of CC4348. Before the treatment (0 min), the abundances of these transcripts were similar in CC4348 and *gpx5*, but they increased significantly after addition of RB in L27, and the increase was between that of CC4348 and *gpx5* cells. This increase suggested that ROS stimulated the upregulation of transcripts encoding proteasome-related subunits. As ROS concentration increased, protein stress increased, resulting in upregulation of the ubiquitin–proteasome system.

Oxidative stress also induced 11 transcripts encoding proteins in the secretory pathway or vacuolar assembly/sorting proteins (*VPS2A*, *VPS26*, *VPS46*, and *VPS60*) from subgroup B1. These transcripts increased in abundance specially in the *gpx5* mutant after RB treatment. These data suggest that high ROS levels accelerated the transport and exchange of metabolites between cells.

Transcripts from subgroup B3 genes were specifically downregulated in the *gpx5* mutant, including ten adenylate cyclase or guanylate cyclase subunit transcripts and five 3’,5’-cyclic nucleotide phosphodiesterase transcripts, whereas there was no significant difference in CC4348. Adenylate cyclase and guanylate cyclase catalyze the formation of cAMP and cGMP, respectively. As secondary messengers, cAMP or cGMP activate kinases. The high concentration of ROS could inhibit some cAMP or cGMP signaling pathways. In addition, subgroup B3 included transcripts encoding Mitogen-Activated Protein Kinase (MAPK) signaling pathway factors. For example, *MAPKKK7* (Mitogen-Activated Protein Kinase Kinase Kinase) was specifically downregulated, from 19 to 9 RPKM in the *gpx5* mutant, but its expression remained stable at about 16 RPKM in CC4348. ROS also affected the calcium signal as mediated by calmodulin. Transcripts encoding calmodulins (Cre03.g210177 and Cre03.g170200), a Ca^2+^-transporting ATPase (Cre12.g505350), and a calcium-binding protein (Cre16.g680500) from subgroup B3 were specially downregulated in the *gpx5* mutant but showed no significant difference in CC4348. This suggests that signal transduction including cAMP or cGMP, MAPK pathways, and Ca^2+^ signaling were inhibited in response to high concentrations of ROS.

Two iron transporter genes, *IRT2*, encoding an iron-responsive ZIP family transporter (Cre12.g530350) and *FTR1*, encoding an iron transporter (Cre03.g192050), responded to ^1^O_2_ specially in the *gpx5* mutant. We speculated that iron might respond to the cellular redox state or deliver a redox signal to other ROS sensors. 

Similarly, group C genes, which were downregulated in CC4348, were classified into subgroups. Interestingly, none of the genes were upregulated in the *gpx5* mutant (group C1); 33 genes, designated C2, were not significantly changed in the *gpx5* mutant; and 445 genes, designated C3, were downregulated in both *gpx5* and CC4348 (Figure 5C). 

The C2 genes encoded proteins involved in nucleotide metabolism and regulation of transcription. Transcripts of a histone H2B variant (*HBV1*), ATP-dependent RNA helicases (*HEL12*, *HEL34*, and *HEL56*), DNA-directed RNA polymerase I subunit (*RPA12*), and transcription initiation factor TFIID subunit 10 (Cre16.g657000) were downregulated more than 2-fold in CC4348, but remained stable in the *gpx5* mutant under oxidative stress. These downregulated transcripts included some transcription factors or transcription regulators, including protein with chromosome condensation (*RCC1*) repeat domain (Cre12.g528350), ribosomal N-lysine methyltransferase 3 (Cre12.g541777), and leucine zipper transcription factors (Cre06.g252000 and Cre12.g490950). These transcription factors were specifically inhibited in CC4348, which might be related to GPX5 protein or low ROS concentrations.

The expression of subgroup C3 genes was downregulated both in the *gpx5* mutant and CC4348 under oxidative stress. In subgroup C3, 178 FAP transcripts were rapidly downregulated under oxidative stress in the *gpx5* mutant and CC348. These results further indicated that oxidative stress inhibits the expression of flagella-related genes, and more ROS induced deflagellation in *C. reinhardtii* (Figure 3D). In addition, the transcripts of four genes encoding selenoproteins, *GPX1*, *NTR1*, *SELW1*, and *SELU1*, from subgroup C3 were significantly downregulated in CC4348 and *gpx5* cells under oxidative stress. The relationship between selenoproteins and oxidative stress remains to be explored.

By comparing the expression of these 1852 DEGs in CC4348 and L27, we found that there were high similar responses between CC4348 and the complementary strain L27. Among 322 upregulated DEGs in CC4348 after treatment for 30 min, roughly 79% (253) overlapped with induced DEGs in L27. Among 386 downregulated DEGs in CC4348 after treatment for 30 min, roughly 88% (341) overlapped with depressed DEGs in L27. Overall, about 84% (594 out of 708) of DEGs in CC4348 were overlapped with that in L27 after treatment for 30 min (Appendix A).

In brief, after RB treatment, CC4348 and the *gpx5* mutant shared some common pathways to respond to oxidative stress showed, including upregulated ROS detoxification and ubiquitin-mediated proteolysis (subgroup A1), and inhibited the expression of cell structure related proteins and selenoproteins (subgroup C3). When GPX5 protein was deficient, some specific responses occurred, such as accelerated the TCA cycle and mETC in mitochondria to supply more ATP (subgroup B1), repressed genes related to chlorophyll metabolism and photosynthesis (subgroup A2 and A3), and downregulated iron transporters (subgroup B3). 

### 3.5. Effect of ^1^O_2_ on Photosynthesis

Previous studies on terrestrial plants and photosynthetic microorganisms have shown that hydrogen peroxide downregulates the expression of genes encoding photosynthesis-related proteins [45,46,47]. To further understand the effects of ^1^O_2_ on the expression of photosynthesis-related genes, we analyzed the transcript changes of LHC genes. The transcript level of *LHCs* in CC4348 increased by two- to four-fold within 1 h after RB treatment. However, they fluctuated over a two-fold range during oxidative stress in the *gpx5* mutant, but increased significantly in L27, even more than in CC4348 after RB treatment for 30 min (Figure 6A–C and Appendix A).

RNA-seq data of transcripts encoding proteins involved in chlorophyll metabolism in CC4348, *gpx5*, and L27 strains were also analyzed at 0, 30, and 60 min after RB treatment (Figure 6D–F). The expression of chlorophyll-metabolism-related genes in CC4348 showed a continuous trend of upregulation within 60 min of oxidative stress, whereas those in the *gpx5* mutant showed an upward trend in expression within the first 30 min and remained stable in the following 30 min, except a chlorophyll synthase gene (*CHLG*, Cre06.g294750) and a delta-aminolevulinic acid dehydratase gene (*ALAD1*, Cre02.g091050) (Figure 6E). Most of these transcripts were upregulated by less than 6-fold after 30 min and then remained stable abundance in L27. In CC4348, the transcript abundance of *CHLG* was continuously increased from 12 RPKM after RB treatment and reached a peak of 45 RPKM after 60 min; in the *gpx5* mutant, *CHLG* reached a peak of 11 RPKM after 30 min and then decreased to 3 RPKM. Although the expression of *CHLG* in the *gpx5* mutant increased by 4.5-fold, the overall transcript abundance was low. CHLG catalyzes the production of chlorophyll from chlorophyllide. The relative low abundance of *CHLG* transcript affect the final synthesis of pigments. In addition, the *ALAD1* transcript increased more than 6-fold (peak at 45 RPKM) after RB treatment for 60 min in CC4348, and increased about 11-fold (peak at 55 RPKM) in *gpx5*. The expression pattern of *ALAD1* in L27 was similar with that in CC4348. *ALAD1* was more strongly induced by ^1^O_2_ than other genes in the chlorophyll synthesis pathway, suggesting that it is important for this response.

To determine the effect of oxidative stress on the photosynthetic efficiency of CC4348 and the *gpx5* mutant, we measured chlorophyll pigment contents and photosynthetic efficiency every 30 min, during 1 h of treatment with 1 µM RB and another 3.5 h after the treatment was terminated. Figure 6G shows the total chlorophyll *a* and *b* contents. Consistent with the mRNA abundance of genes related to chlorophyll metabolism, the chlorophyll content of CC4348 increased by about 7% after a 30-min treatment and then returned to 91% of the initial level after the 3.5 h recovery. The chlorophyll in the *gpx5* mutant increased by only 4% in the first 30 min of treatment but then decreased continuously after removal of RB. The pigment content decreased to 72% of the initial state in the *gpx5* mutant at 3.5 h after termination of the RB treatment. The pattern of change of pigment content in L27 was similar to that of CC348, in which chlorophyll increased by 8% after 30 min of treatment and then decreased to 96% of the initial level. Therefore, we speculated that a low ROS concentration may stimulate the transient synthesis of chlorophyll, whereas a high ROS concentration may inhibit chlorophyll metabolism. 

The maximum efficiency of PSII (Figure 6H) was also measured. The photosynthetic efficiency of PSII averaged around 0.7 in CC4348 throughout the experiment. However, at the initial time point, the maximum efficiency was only 0.63 in the *gpx5* mutant, but this increased to 0.68 after 30 min of RB treatment. At 3 h after the RB was removed, the maximum efficiency of PSII had declined to its lowest value, 0.4, in the *gpx5* mutant. At 3.5 h after the RB was removed, the maximum efficiency slowly recovered to 0.53 in the mutant, which was still lower than its initial level. The maximum efficiency of PSII of L27 was similar to that of CC348, but its value was slightly lower (about 0.6). Oxidative stress had little effect on the photosynthetic efficiency of CC348, but it had a negative effect on the *gpx5* mutant. After treatment with RB, ROS accumulated in *gpx5* cells. Therefore, a high concentration of ROS might inhibit photosynthetic efficiency. After the removal of oxidative stress, the ROS level in cells gradually decreased and the photosynthetic efficiency recovered accordingly. These results indicated that a small amount of ROS had little effect on photosynthetic efficiency, but that a high concentration of ROS inhibited photosynthetic efficiency.

## 4. Discussion

In this study, we performed transcriptomic analyses to explore the function of GPX5 during the acclimatization to ^1^O_2_ generated by RB in *C. reinhardtii*. The different expression pattern of genes in algal strains with or without GPX5 function indicated that cells have specific responses to different levels of stress. Oxidative stress induced the upregulation of genes encoding proteins involved in ROS detoxification and protein degradation in these two strains. However, some genes, such as coproporphyrinogen III oxidase (*CPX1*), were only induced in CC4348. Another set of genes, such as a putative dTDP-glucose 4-6-dehydratase gene (*SNE1*), were only induced in the *gpx5* mutant, in which the ROS level increased more than two-fold. We also investigated the adjustment of the transcriptional network in the *gpx5* mutant at optimized laboratory growth conditions.

### 4.1. Acclimatization of the gpx5 Mutant in Laboratory Growth Conditions

We explored the function of GPX5 under optimized laboratory growth conditions (no extra oxidative stress added). GPX5 deficiency led to a decrease of the ROS-scavenging ability in the mutants (Figure 1A). However, the *gpx5* mutant can still grow under laboratory growth conditions. Functional analysis of DEGs by comparing the transcriptomes in the *gpx5* mutant to CC4348 revealed that the mutant copes with the loss of GPX5 by multiple mechanisms (Appendix A). First, the synthesis of the antioxidant ASC was upregulated in the *gpx5* mutant by increased abundance of the mRNA encoding VTC2, which catalyzes the first, rate-limiting step in ASC synthesis [48,49]. ASC plays an important role in protecting plants from oxidative damage [50,51,52], suggesting that the mutant cells might activate the ASC-dependent detoxification of ROS, thus helping to compensate for the loss of GPX5 function. 

Second, the abundance of transcripts encoding redoxins, such as NRX2, NRX3, and SRX1, increased in the *gpx5* mutant. These redoxins reduce disulfide bonds in proteins, to relieve cellular oxidative stress. In addition, chlorophyll synthesis declined, reducing the amount of ROS formation. In brief, under optimized laboratory growth conditions, the *gpx5* mutant displayed enhanced production of antioxidants and redoxins, as well as weakened ROS production pathways to compensate for the lack of GPX5 (Appendix A). 

### 4.2. Adjustment of Photosystems in Response to Oxidative Stress

Under optimized laboratory growth conditions, the transcript abundance of most of the genes encoding chlorophyll metabolism and photosynthetic antenna proteins were lower in the *gpx5* mutant than in CC4348 (Appendix A). Consistent with these mRNA levels, the chlorophyll content in *gpx5* mutant was lower than that in CC4348 (Figure 6G). Chlorophyll is another photosensitizer; therefore, the reduction of chlorophyll contents might protect cells from more oxidative stress. After treatment with RB for 60 min, chlorophyll-synthesis-related transcripts were upregulated, consistent with the increased content of chlorophyll after treatment in CC4348. Nevertheless, the expression of chlorophyll-synthesis-related transcripts fluctuated within 2-fold in the *gpx5* mutant. The chlorophyll content of the *gpx5* cells increased slightly during the first 30 min of treatment, but the increase was lower than that of CC4348. However, it decreased in the next 30 min of treatment and continued to decline for 3.5 h after RB was removed (Figure 6D,G). Since high light levels lead to robust photosynthesis but also ROS generation, the logical response of the cell is to increase chlorophyll, to better absorb the available light. When ROS is too high, photosynthesis becomes a net damaging process and must be scaled back. In a similar way, under treatment with RB, increasing ROS levels facilitated chlorophyll metabolism to physically dissipate excessive energy. When increasing ROS overwhelm the ability to detoxify, photosynthesis would be downregulated. These results suggest that short-term low levels of ROS induce a transient increase in the metabolism of chlorophyll, to better absorb the available light, but high or sustained level of ROS inhibit chlorophyll synthesis; therefore, photosynthesis becomes a net damaging process and must be scaled back.

### 4.3. Regulation of Iron Homeostasis in Response to Oxidative Stress 

Among 708 DEGs, *ferritin2* exhibited the highest transcript abundance, increasing 150-fold in CC4348 after a 1 h treatment with RB (Figure 3B,C) and increasing 338-fold in the *gpx5* mutant after a 1 h treatment with RB. This upregulation of *ferritin2* was consistent with a previous study, in which the expression of *ferritin2* increased about 80-fold in response to RB and 10-fold in response to H_2_O_2_ [17,44]. By contrast, in the *sak1* mutant, the transcript abundance of *ferritin2* was extremely low after RB treatment (<10 RPKM) [17]. Ferritin2 is highly similar to bacterial DPS (DNA-binding protein from starved cells), which protects DNA from oxidative damage by sequestering Fe ions [53,54]. These data suggested that Fe-catalyzed DNA damage caused by ^1^O_2_ is more serious than that caused by H_2_O_2_, especially in cells that lack GPX5_._ Besides *ferritin2*, transcript abundance for another two Fe-containing protein genes, *SOUL1* and *SOUL2*, increased by more than 60-fold in CC4348 and more than 120-fold in the *gpx5* mutant after RB treatment for 30 min. Increased levels of Fe-containing proteins would chelate more Fe to reduce Fe-catalyzed DNA damage.

In addition to Fe chelation, Fe mobilization and Fe assimilation played important roles in Fe homeostasis. *Chlamydomonas* has two Fe-uptake pathways, the high-affinity pathway, involving the ferroxidase FOX1 coupled to a ferric ion transporter FTR1, and the lower-affinity pathway, involving inducible ZIP family transporters IRT1 and IRT2 [55,56]. Iron assimilation components also include ferrireductase (FRE1) and Fe-assimilating protein FEA1 (Cre12.g546550). FEA1 is a secreted protein that facilitates high-affinity iron uptake, perhaps by concentrating iron in the vicinity of the cell [55]. The transcripts of all iron mobilization and iron assimilation components, including *FOX1*, *FTR1*, *IRT2*, and *FRE1*, did not significantly change in CC4348, except *FEA1*, which was upregulated about 3-fold after RB treatment for 1 h. However, all of these components decreased in transcript abundance in the *gpx5* mutant. These meant that lowering the concentration of Fe by chelation and mobilization likely enhances the ability of *Chlamydomonas* to tolerate oxidative stress.

### 4.4. The Effect of ^1^O_2_ on the Expression of Selenoproteins 

Although selenocysteine has stronger reducibility than cysteine, selenoproteins with selenocysteine in the active site were not all induced by oxidative stress in our study. Twelve transcripts of selenoproteins were detected in *C. reinhardtii* cells, including nine downregulated and three upregulated transcripts (Appendix A). Among them, two selenocysteine glutathione peroxidases, GPX1 and GPX2, are involved in the degradation of hydrogen peroxide and belong to the same family as GPX5. However, these two transcripts showed an unexpected downregulation, in contrast to that of *GPX5*, in both strains. Although not significantly changed, the transcript abundance of *GPX2* decreased slightly in CC4348 and the *gpx5* mutant. NADPH-dependent thioredoxin reductase (NTR) has been reported to play a role in abiotic stress responses in *Arabidopsis* [57]. In CC4348 cells, the transcripts encoding selenium-containing NTR1 (Cre08.g368400) were downregulated, while transcripts of the gene encoding selenium-free NTR2 (Cre02.g098850) were upregulated after the addition of RB. In addition, the transcript abundance of the selenium-binding protein gene *SBD1* (Cre03.g166050) significantly increased under oxidative stress in both CC4348 and *gpx5* cells. The *Lotus japonicus* homolog of SBD1 is thought to have more than one physiological role and has been implicated in controlling the oxidation/reduction status of target proteins in vesicular Golgi transport [58]. In summary, oxidative stress influenced the expression of some selenoproteins, but the underlying mechanism remains to be further studied.

In addition to members of the GPX and NTR families, the members of other protein families showed different expression patterns during oxidative stress. For example, *NRX1* did not significantly change in CC4348, but it increased in *gpx5*, in contrast to the pattern of *NRX2*. *TRX21* and *SELU1* from the thioredoxin superfamily were downregulated in CC4348 and *gpx5* under oxidative stress, while another 8 members were upregulated (*LCI7*, *PDI1*, *PDI2*, *PDI4*, *PDIA6*, *TRX5*, *NRX2*, and *Endoplasmic Reticulum-Golgi Intermediate Compartment* (*ERGIC*, Cre08.g358579)). These observations indicated that even members of the same protein family function independently.

### 4.5. GPX5 is Necessary for ^1^O_2_ Acclimation in C. reinhardtii

Two proteins responsible for the response to ^1^O_2_, the C2H2 zinc finger protein MBS (Cre09.g416500) [16] and the putative transcription factor SAK1 (Cre17.g741300) [17], are located in the cytosol, as is the GPX5 protein [15]. Therefore, we wondered about the relationship among these three proteins in ^1^O_2_ signaling. The ^1^O_2_-responsive reporter gene (*HPS70A*) was not induced by RB in the *mbs* or the *sak1* mutant. In the *gpx5* mutant, the transcript abundance of *HSP70A* increased by 12-fold compared with CC4348. So, SAK1 and MBS might function in the same signaling pathway, but GPX5 does not function in this pathway. Furthermore, the expression of *MBS* was not affected after RB treatment in the *sak1* mutant, so the MBS protein likely functions upstream of SAK1 in the response to ^1^O_2_.

The transcriptome of CC348 after treatment with 1 µM RB for 1 h was compared to the transcriptome of 4A+, the parental strain of the *sak1* mutant, with the same treatment. Among 402 upregulated genes in CC4348, 95 were also upregulated in 4A+. These transcripts encoded redox-related proteins and ATP-binding cassette (ABC) transporters, as well as GPX5 and SAK1. *GPX5* was upregulated 17-fold in CC4348 and 14-fold in 4A+ after treatment with 1 µM RB. Similarly, *SAK1* was upregulated 6-fold in CC4348 and 8-fold in 4A+ in the same conditions. Some other strongly ^1^O_2_-responsive genes also showed the same pattern, such as *SOUL1*, *SOUL2*, *CFA1*, *CFA2*, *LCI7*, and *VTC2*. These data suggest that our transcriptome data are reliable.

Compared the transcriptomics data from the *gpx5* mutant and the *sak1* mutant subjected to oxidative stress treatment using 1 µM RB for 1 h, there were 456 overlap DEGs which were induced in both the *gpx5* mutant and the *sak1* mutant, while 106 overlap DEGs depressed. When *C. reinhardtii* cells were treated with RB, the TCA cycle was induced by oxidative stress in the *gpx5* mutant, whereas these changes were absent in the *sak1* mutant. We also detected numerous transcripts encoding cell structure proteins, including FAPs, which were downregulated after RB treatment in the *gpx5* mutant. However, some of them increased in abundance in the *sak1* mutant after RB treatment. For example, the transcripts encoding alpha tubulin (*TUA1* and *TUA2*) and beta tubulin (*TUB1* and *TUB2*) were downregulated in CC348 and *gpx5* after RB treatment, but they showed no significant change and high abundances (>500 RPKM). Other factors might affect cell structure proteins besides oxidative stress. 

We identified the SAK1 transcript in the *gpx5*, control, and complemented strains. After treatment with 1 µM RB for 1 h, the SAK1 transcript increased in abundance, from 6 to 36 RPKM in CC4348, from 6 to 107 RPKM in the *gpx5* mutant, and from 14 to 179 RPKM in complemented strain L27. We also found the GPX5 transcript in the *sak1* mutant from the published data [17], which increased in abundance, from 314 to 1112 RPKM. These data indicated that SAK1 protein and GPX5 protein did not function in the same pathway, and the expression of cell structure proteins may be regulated separately. We also observed similarities and differences in the response to ^1^O_2_ and H_2_O_2_. After RB treatment, 18 transcripts involved in chlorophyll biosynthesis were upregulated in CC4348, most of which remained unchanged in the *gpx5* mutant and *sak1* mutant. By contrast, almost all of these were downregulated after H_2_O_2_ treatment [44]. During 1 h of RB treatment, the increased photosystem transcripts did not have enough time to elevate the photosynthetic efficiency in CC4348. Transiently increased photosynthetic efficiency was observed in the *gpx5* mutant. After cells were treated with H_2_O_2_, the photosynthetic efficiency slowed down in the first 0.5 h and returned after 4 h. Maybe the initial burst of ^1^O_2_ stimulated the photosystem, while H_2_O_2_ had a negative impact on photosynthetic rate. Therefore, ^1^O_2_ and H_2_O_2_ affect cell physiology in different ways.

### 4.6. The Genome-Wide Response to ^1^O_2_ in gpx5 Mutants

By comparing the transcriptomes of the *gpx5* mutant and CC348 after RB treatment, we examined the specific responses to different levels of oxidative stress. When CC4348 was treated with RB, the ROS level increased only 1-fold. Under this condition, antioxidant enzymes, such as GPX family proteins, and the biosynthesis of the antioxidant ASC were induced. The chlorophyll contents increased and carbon metabolism, including glycolysis and the OPP pathway, were upregulated. The abundance of ferritin2 was strongly increased to chelate more iron, thus protecting biomacromolecules from oxidation by iron. At the same time, the iron transporters were unchanged in abundance, and FEA1 was upregulated, to avoid introducing extracellular iron into cells. In addition, transcripts encoding selenoproteins and cell structural proteins, especially FAPs, were significantly downregulated under oxidative stress (Figure 7A). 

Cells lacking GPX5 were in severe oxidative stress after being subjected to RB treatment, and their ROS levels increased up to 3-fold. Under this condition, the GPX5-mediated pathway to detoxify into H_2_O_2_ was defective, resulting in high levels of ROS. In addition to the shared responses with CC4348 after RB treatment, some specific responses occurred in the *gpx5* mutant. Continuous high levels of ROS repressed genes related to chlorophyll metabolism and photosynthesis. Serious oxidative stress accelerated the TCA cycle and mETC in mitochondria to supply more ATP. Ferritin2 was also induced, and iron transporters were downregulated, to enhance the protection from damage by iron. However, *FEA1* was downregulated. At high levels of ROS, *SAK1* was still increased in abundance, but *MBS* decreased in the *gpx5* mutant after treatment. By using multiple adaption mechanisms, cells managed to survive and recover from oxidative stress (Figure 7B).

In summary, this study revealed that GPX5 plays a crucial role in maintaining ROS homeostasis in cells and mediating the dose-dependent response to ROS stress in *Chlamydomonas*. The function of GPX5 cannot be replaced completely by other ROS-scavenging enzymes or systems. Iron homeostasis may play an important role in acclimatization to oxidative stress.

## Figures and Tables

**Figure 1 genes-11-00463-f001:**
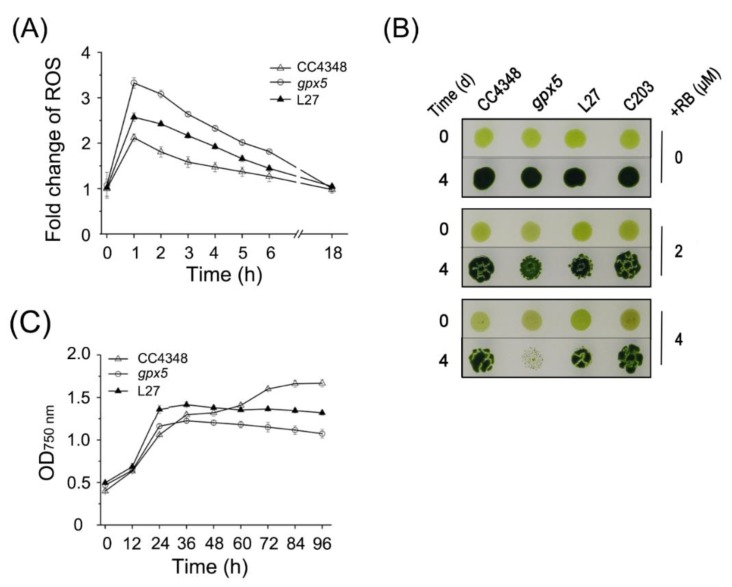
GPX5 maintains ROS homeostasis in *Chlamydomonas reinhardtii*. (**A**) Intracellular ROS concentrations in the cells of CC4348, the *gpx5* mutant, and the rescued strain L27 treated with 2 μM rose bengal (RB) for 1 h. (**B**) The effect on the growth of CC4348, the *gpx5* mutant, and the complemented lines L27 and C203 treated with 2 or 4 μM RB for 1 h. (**C**) Growth curves of CC4348, the *gpx5* mutant, and L27 cells under laboratory growth conditions. The error bars represent standard deviations of three biological replicates.

**Figure 2 genes-11-00463-f002:**
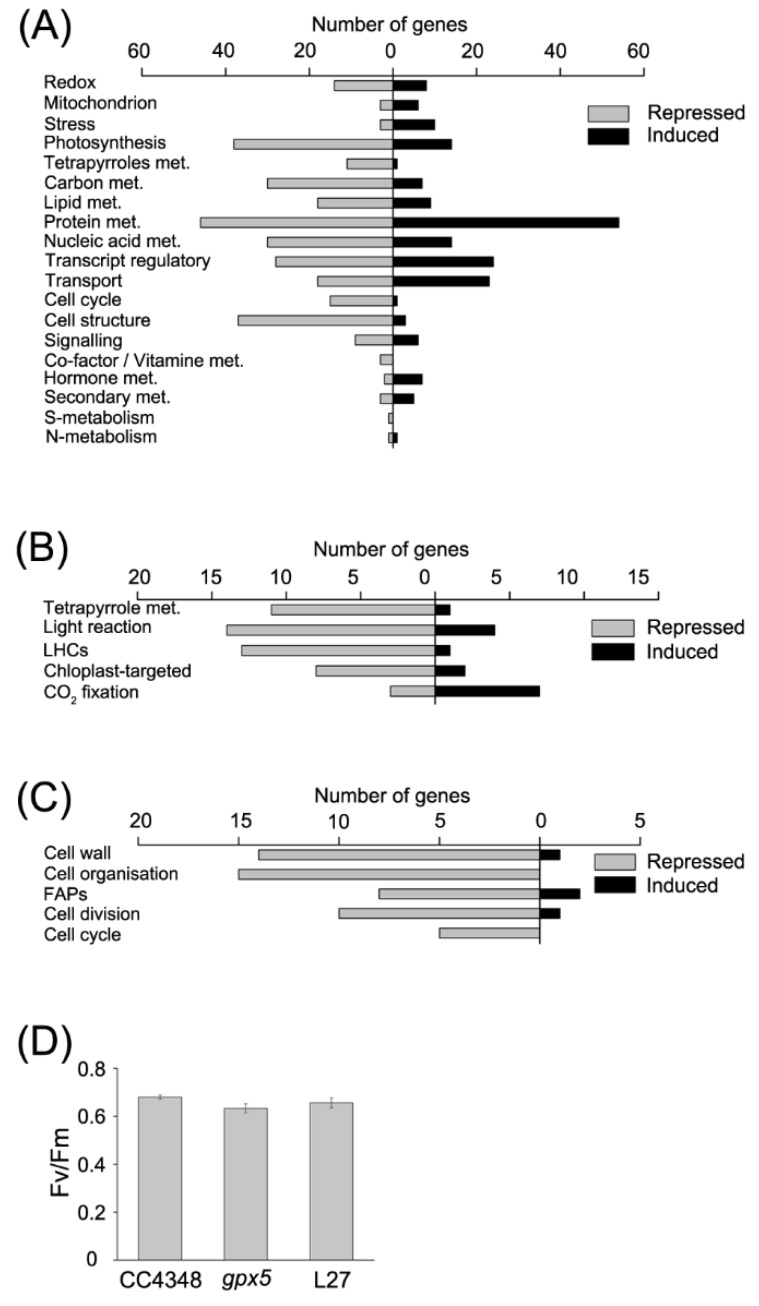
Functional analysis of differentially expressed genes in the *gpx5* mutant compared to the parental strain CC4348 under laboratory growth conditions. (**A**) Functional subcategories of differentially expressed genes (DEGs) in the *gpx5* mutant compared to the parental strain CC4348 under laboratory growth conditions. For clarity, only the genes with assigned functions are shown. Functional analysis of DEGs passing the criteria fold change greater than two (up) or less than 0.5 (down) with FDR < 1%, based on the MapMan functional classes; met, metabolism. (**B**) List of the DEGs related to photosynthesis and tetrapyrrole metabolism in the *gpx5* mutant compared to the parent strain CC4348 under laboratory growth conditions. LHC, light-harvesting complex. (**C**) List of the DEGs related to cell structure and the cell cycle in the *gpx5* mutant compared to the parental strain CC4348 under laboratory growth conditions. FAPs, flagellar-associated proteins. (**D**) The maximum efficiency of PSII (Fv/Fm) of CC4348, the *gpx5* mutant, and L27 cells at a density of 2 × 10^6^ cells mL^−1^ in exponential phase under laboratory growth conditions. The error bars in the graph represent standard deviations of three biological replicates.

**Figure 3 genes-11-00463-f003:**
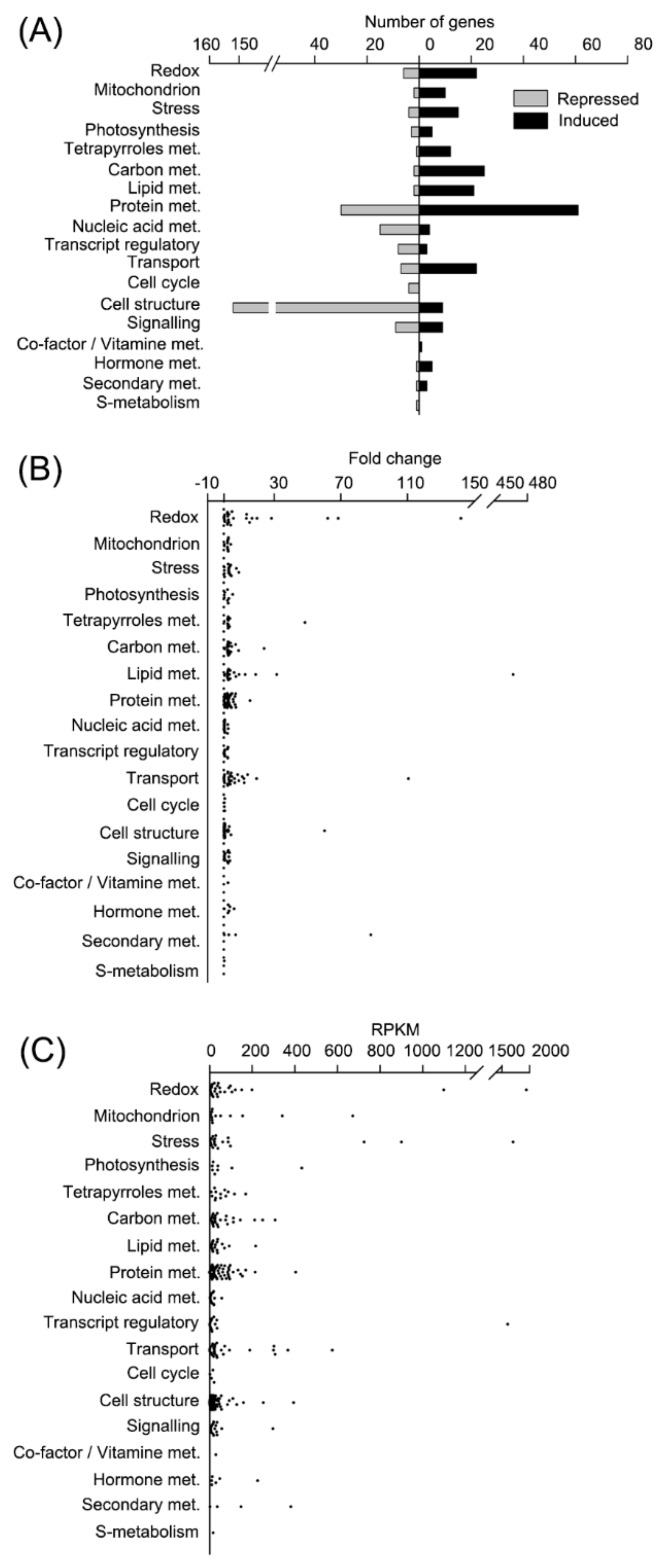
Functional analysis of differentially expressed genes in CC4348 after being treated with 1 μM RB for 30 min. (**A**) Functional subcategory of differentially expressed genes (DEGs) in CC4348 treated with 1 μM RB for 30 min. For clarity, only the genes with assigned functions are shown. Functional analysis of DEGs passing the criteria of fold change greater than two (up) or smaller than 0.5 (down) with FDR < 1%, based on the MapMan functional classes; met, metabolism. (**B**) The fold change of transcript abundance of DEGs in (**A**). The X-axis indicated the logarithm based two of the ratio of transcript abundance of genes after treated with 1 μM RB for 30 min to control in CC4348. (**C**) The relative transcript abundance of DEGs in (**A**). The X-axis indicated the value in unit of RPKM of transcript abundance after treated with 1 μM RB for 30 min in CC4348.

**Figure 4 genes-11-00463-f004:**
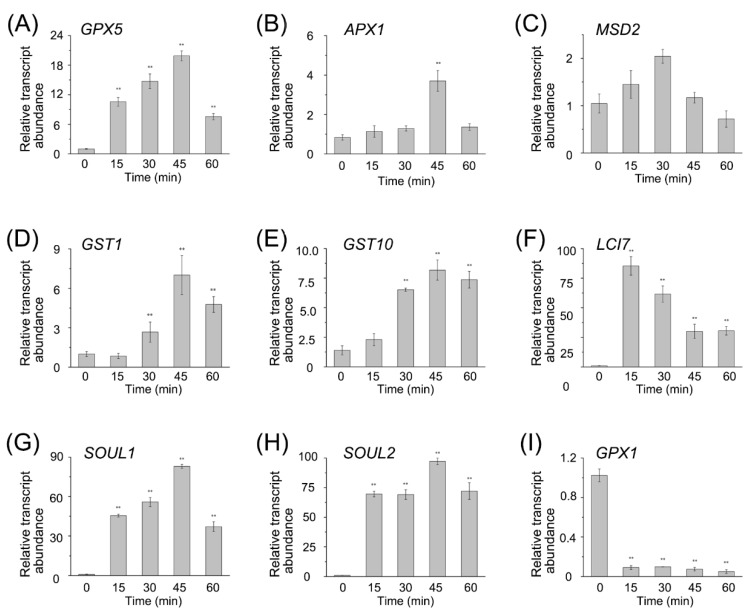
Transcriptional changes of antioxidant-related genes in RB-treated CC4348 cells. The transcript abundance of *GPX5* (**A**), *APX1* (**B**), *MSD2* (**C**), *GST1* (**D**), *GST10* (**E**), *LCI7* (**F**), *SOUL1* (**G**), and *SOUL2* (**H**) and *GPX1* (**I**) at 15, 30, 45, and 60 min after treatment with 1 μM RB in CC4348 were detected, using qRT-PCR. *CβLP* was used as an internal control to normalize the data. The error bars indicate standard deviation of biological triplicates. The asterisks represent statistical significance between every treated group and the control, which was determined by *t*-test (*n* = 3, *P* < 0.01).

**Figure 5 genes-11-00463-f005:**
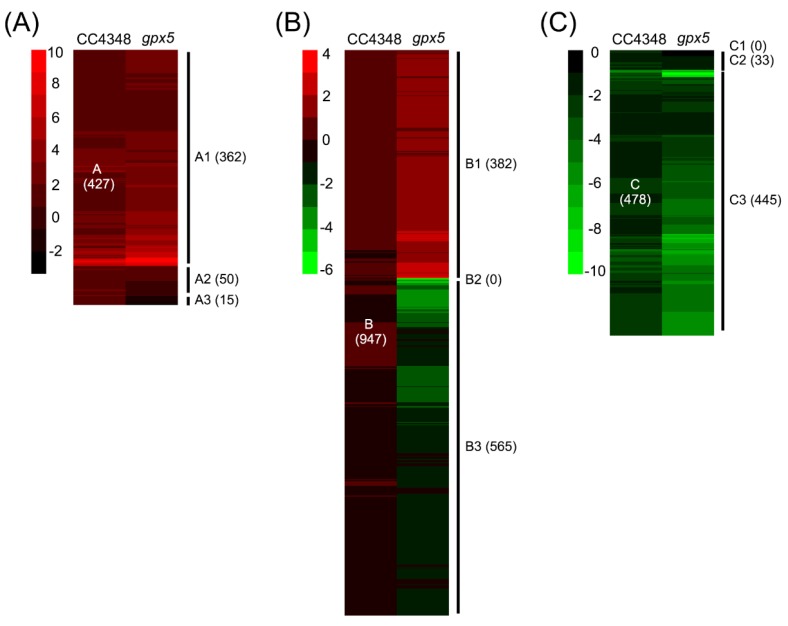
Comparison of the transcriptional response of CC4348 versus the *gpx5* mutant after treatment with 1 μM RB for 30 min. The cluster diagrams depict the fold change in transcript levels determined by RNA-seq. The log_2_ relative expression values of 1 and −1 were selected as thresholds to designate the categories of transcripts that accumulated and declined, respectively. (**A**) The upregulated DEGs in CC4348 were categorized into **A**1 (upregulated in the *gpx5* mutant), **A**2 (no significant change in the *gpx5* mutant), and **A**3 (down regulated in the *gpx5* mutant). (**B**) The genes without significant changes in CC4348 but with significant changes in the *gpx5* mutant were categorized into **B**1 (upregulated in the *gpx5* mutant), **B**2 (no significant change in the *gpx5* mutant), and **B**3 (downregulated in the *gpx5* mutant). (**C**) The downregulated DEGs in CC4348 were categorized into **C**1 (upregulated in the *gpx5* mutant), **C**2 (no significant change in the *gpx5* mutant), and **C**3 (downregulated in the *gpx5* mutant).

**Figure 6 genes-11-00463-f006:**
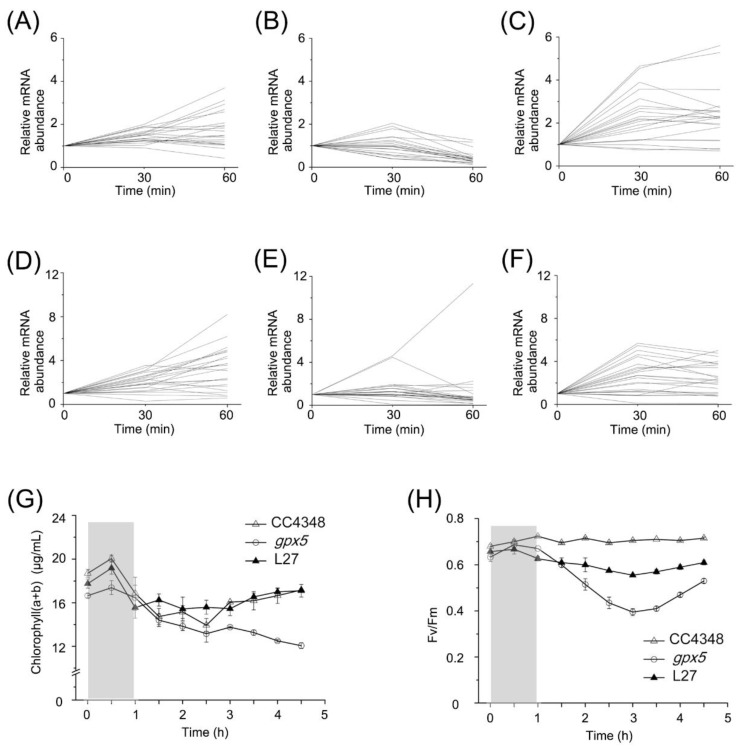
The response of photosystem to ^1^O_2_. (**A**–**C**) The expression pattern of DEGs encoding light-harvesting proteins in CC4348 (**A**), the *gpx5* mutant (**B**), and L27 (**C**) after treated with 1 μM RB determined by RNA-seq. (**D**–**F**) The expression pattern of DEGs related to chlorophyll metabolism in CC4348 (**D**), the *gpx5* mutant (**E**), and L27 (**F**) after treated with 1 μM RB determined by RNA-seq. The initial RPKM was set as 1 for each strain, separately, to normalize the data. Genes and corresponding RPKM values used in this analysis can be found in Appendix A. (**G**,**H**) Comparison of the content of chlorophyll (*a* + *b*) (**G**) and maximum efficiency (Fv/Fm) of photosystem II (**H**) in CC4348, the *gpx5* mutant and L27. The same amounts of cells were treated with 1 μM RB for 1 h, and the RB was then washed out, and measurements were taken over a period of 3.5 h. The stage of treatment is shaded with light gray.

**Figure 7 genes-11-00463-f007:**
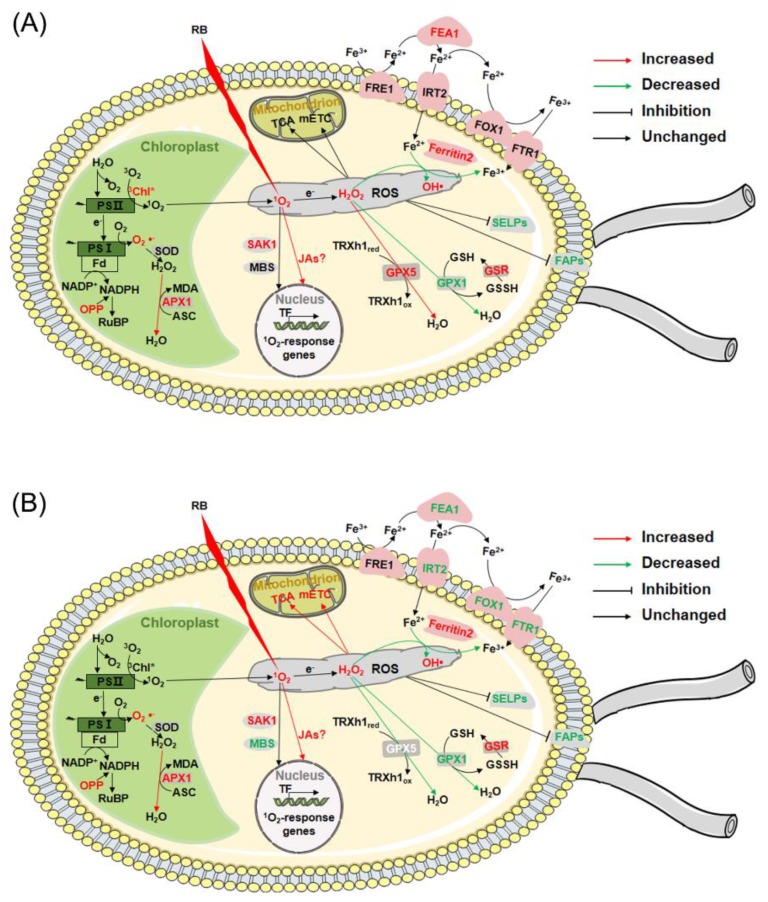
Summary of the cellular response to oxidative stress in CC4348 (**A**) and the *gpx5* mutant (**B**). (**A**) After treatment with RB, the CC348 cells upregulated the ROS detoxification system and inhibited the expression of flagellar associated proteins (FAPs) and selenoproteins (SELPs). (**B**) When the *gpx5* mutant suffered the treatment with RB, expect for the same responses of ROS detoxification system, FAPs and SELPs, several specific responses occurred, such as accelerated the TCA cycle and mETC in mitochondria, repressed genes related to chlorophyll metabolism and photosynthesis, and downregulated iron transporters. Red characters indicate upregulated expression of related proteins or metabolic pathways after RB treatment, while green characters indicate downregulated expression of related proteins or metabolic pathways after RB treatment. Abbreviations: APX, ascorbate peroxidase; ASC, ascorbate; FAPs, flagellar associated proteins; Fd, ferredoxin; FEA1, Fe-assimilating protein 1; FOX1, ferroxidase; FRE, ferrireductase 1; FTR1, ferric ion transporter 1; GPX, glutathione peroxidase; GSR, glutathione reductase; GSH, glutathione; GSSG, oxidized glutathione; IRT2, ferrous ion transporter 2; mETC, mitochondrial electron transfer chain; OPPP, oxidative pentose phosphate pathway; ROS, reactive oxygen species; PRX, peroxiredoxin; PSI/II, photosystem I/II; RB, rose bengal; RuBP, ribulose 1,5-bisphosphate; SELPs, selenoproteins; SOD, superoxide dismutase; TF, transcription factor; TRXh1, thioredoxin h1. SAK1 [17] and MBS [16] are two cytosolic proteins required for induction of nuclear gene expression responses by ^1^O_2_.

**Table 1 genes-11-00463-t001:** Overview of metabolic pathways and cellular processes in each subgroup from Figure 5. The numbers in brackets represent the counts of according subgroups.

Specific in (CC4348 + RB) Low ROS	Shared	Specific in (*gpx5* + RB) High ROS
Induced		
A2+A3 (50+15)	A1 (362)	B1 (382)
Chlorophyll metabolism,	^a^ ROS detoxification,	Mitochondrial electron transport,
Transport ATPases,	Ubiquitin-mediated proteolysis,	TCA cycle,
Transcript regulatory,	Glycolysis, OPP,	^b^ Ubiquitin-mediated proteolysis,
FEA1.	Ferritin2.	Secretory pathway.
Repressed		
C2 (33)	C3 (445)	B3 (565)
Nucleotide metabolism,	^c^ Cell structure proteins,	Signal transduction,
Transcript regulatory.	Selenoproteins.	Iron transporters.

(^a^ Some transcripts were also found in class B1. ^b^ Some transcripts were also found in class A1. ^c^ Some transcripts were also found in class B3. FEA1, Fe-assimilating protein; Ferritin2, ferritin-like protein.)

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
