# Peer review of "Transcriptomic and Physiological Responses to Oxidative Stress in a Chlamydomonas reinhardtii Glutathione Peroxidase Mutant"

_genes, 2020, doi:10.3390/genes11040463_

Round 1

Reviewer 1 Report

This paper consists of a thorough transcriptomic analysis of the effect of rose bengal oxidative stress on Chlamydomonas, and the impact of a mutation in gpx5 on those responses.  Transcriptomic studies rarely can prove anything about biological mechanisms as such, but they are a fertile way to generate hypotheses and suggest what is going on in a cell under specific conditions. The authors provide a cogent and convincing set of interpretations of the data which should point the way to future mechanistic studies. The data have been generated using well established approaches and appear to be of good quality, and the figures are well presented. The flow of the paper is logical and the standard of language is good.  I have attached an annotated document with a few minor comments and some minor changes highlighted. 

Author Response

Comments and Suggestions for Authors

This paper consists of a thorough transcriptomic analysis of the effect of rose bengal oxidative stress on Chlamydomonas, and the impact of a mutation in gpx5 on those responses. Transcriptomic studies rarely can prove anything about biological mechanisms as such, but they are a fertile way to generate hypotheses and suggest what is going on in a cell under specific conditions. The authors provide a cogent and convincing set of interpretations of the data which should point the way to future mechanistic studies. The data have been generated using well established approaches and appear to be of good quality, and the figures are well presented. The flow of the paper is logical and the standard of language is good. I have attached an annotated document with a few minor comments and some minor changes highlighted.

Thanks for the reviewer`s positive comments sincerely.

Point 1: Lines 95-98: The nonpolar, nonionic DCFH-DA that becomes nonfluorescent DCFH after crossing cell membranes and flowing enzymatically hydrolyzed by intracellular esterases. Hydrogen peroxide oxidized DCFH to fluorescent dichlorofluorescein (DCF). The ROS abundance after treatment with RB was calculated using the equation: (FTAP+RB)/FTAP.

Response 1:

(1) delated the word that after the word DCFH-DA;

(2) revised the word flowing to being;

(3) revised the word oxidized to oxidizes;

(4) revised the word calculated to quantified.

Revised as follows:

The nonpolar, nonionic DCFH-DA becomes nonfluorescent DCFH after crossing cell membranes and being enzymatically hydrolyzed by intracellular esterases. Hydrogen peroxide oxidizes DCFH to fluorescent dichlorofluorescein (DCF). The ROS abundance after treatment with RB was quantified using the equation: (FTAP+RB)/FTAP.

Point 2: Lines 100-102: Ten-milliliter sample cultures were collected by centrifugation for 3 min at 2500 g and 4°C, washed in RNase-free water, and resuspended in 5 mL of Trizol reagent (Invitrogen). The suspension 1O2 was immediately rocked for 5 min before it was flash-frozen in liquid nitrogen and stored at -80°C.

Response 2:

(1) revised the word milliter to mL;

(2) deleted the word immediately.

Revised as follows:

Ten mL sample cultures were collected by centrifugation for 3 min at 2500 g and 4°C, washed in RNase-free water, and resuspended in 5 mL of Trizol reagent (Invitrogen). The suspension 1O2 was rocked for 5 min before it was flash-frozen in liquid nitrogen and stored at -80°C.

Point 3: Line 124: RNA was from samples after treatment with 1 µM RB immediately (0 min) and after 15, 30, 45, and 60 min.

Response 3:

Omit the verb ‘isolated’.

Revised as follows:

RNA was isolated from samples after treatment with 1 µM RB immediately (0 min) and after 15, 30, 45, and 60 min.

Point 4: Line 132: Photosynthetic efficiency was measured as the PSII maximum quantum yield with Fv/Fm parameter.

Response 4:

Revised as follows:

Photosynthetic efficiency was measured as the PSII maximum quantum yield using the Fv/Fm parameter.

Point 5: Line 134: Three milliliters of culture at a density of 1 x 106 cells/mL was collected and dark-adapted for 15 min. Fv/Fm was measured using a Multiple excitation wavelength modulated chlorophyll fluorometer (Heinz Walz Gmbh Effeltrich, Germany).

Response 5:

Revised as follows:

Three mL of culture at a density of 1 x 106 cells/mL was collected and dark-adapted for 15 min. Fv/Fm was measured using a multiple excitation wavelength modulated chlorophyll fluorometer (Heinz Walz Gmbh Effeltrich, Germany).

Point 6: Lines 138-139: The precipitation was dissolved into 1 mL of methyl alcohol for 24 h in darkness at 4°C.

Response 6:

(1) The word ‘precipitation’ could be replaced by ‘precipitate’;

(2) The phrase ‘dissolved in’ would be more suitable than ‘dissolved into’;

(3) The phrase ‘methyl alcohol’ should be revised to ‘methanol’.

Revised as follows:

The precipitate was dissolved in 1 mL of methanol for 24 h in darkness at 4°C.

Point 7: Line 183: In order to quantify the effect of GPX5 to growth without RB treatment

Response 7:

It should be used the preposition ‘on’ following the verb ‘effect’.

Revised as follows:

In order to quantify the effect of GPX5 on growth without RB treatment

Point 8: Line 218: These data suggest that other GPX proteins did not compensate for the lack of GPX5.

Response 8: These data suggested that the expression of other GPX proteins were not upregulated to compensate for the lack of GPX5.

Point 9: Line 242: These data suggested that some mechanisms of ROS scavenging were enhanced in the gpx5 mutant.

They also suggest that even under unstressed mixotrophic conditions the presence of ROS occurs and is sufficient for cell signalling, even if not for actual cellular stress, implying that signalling is more sensitively detected than stress.

Response 9:

Revised as follows:

These data suggested that some mechanisms of ROS scavenging were enhanced in the gpx5 mutant, and that even under unstressed mixotrophic conditions the presence of ROS occurred and was sufficient for cell signalling.

Point 10: Line 294: These results suggested that the deficiency of GPX5 in the mutant not only affected ROS homeostasis, but also affected cell growth.

It would have been nice to see the effect of the gpx5 mutation on photoautotrophic growth - I expect it would be more dramatic.

Response 10:

It is an interesting point to detect the photoautotrophic growth of the gpx5 mutation. In our previous study, we found that the gpx5 mutation died earlier than the parental strain CC4348 when cultured in nitrogen-free medium. (Miao, R.; Ma, X.; Deng, X., et al., High level of reactive oxygen species inhibits triacylglycerols accumulation in Chlamydomonas reinhardtii. Algal Research 2019, 38, 101400.)

Point 11: Line 301:In summary, intracellular antioxidant metabolism and transport have adjusted to attempt to maintain redox homeostasis in the gpx5 mutant.

Response 11:

inserter the text ‘appear to’.

Revised as follows:

In summary, intracellular antioxidant metabolism and transport appeared to adjust to maintain redox homeostasis in the gpx5 mutant.

Point 12: Lines 659-661: These results suggest that short-term, low levels of ROS induce a transient increase in the metabolism of chlorophyll, but high or sustained level of ROS inhibit chlorophyll synthesis.

I presume the rationale is: since high light levels lead to robust photosynthesis but also ROS generation, the logical response of the cell is to increase chlorophyll to better absorb the available light (as well as state transitions etc). When ROS is too high, photosynthesis becomes a net damaging process and must be scaled back.

Response 12:

Revised as follows:

Since high light levels lead to robust photosynthesis but also ROS generation, the logical response of the cell is to increase chlorophyll to better absorb the available light. When ROS is too high, photosynthesis becomes a net damaging process and must be scaled back. In a similar way, under treatment with RB, increasing ROS levels facilitated chlorophyll metabolism to physically dissipate excessive energy. When increasing ROS overwhelm the ability to detoxify, photosynthesis would be down-regulated.

These results suggest that short-term, low levels of ROS induce a transient increase in the metabolism of chlorophyll to better absorb the available light, but high or sustained level of ROS inhibit chlorophyll synthesis, therefore, photosynthesis becomes a net damaging process and must be scaled back.

Point 13: Line 735-736 Maybe the initial burst of 1O2 irritates the photosystem, while H2O2 has a negative impact on photosynthetic rate.

Response 13: revised the word irritate to stimulate

Revised as follows:

Maybe the initial burst of 1O2 stimulated the photosystem, while H2O2 has a negative impact on photosynthetic rate.

Reviewer 2 Report

The paper contains a considerable amount of work, mainly transcriptomics data, aiming at characterising the gpx5 mutant of Chlamydomonas reinhardtii following oxidative stress. The paper requires improvements. 

line 39: reference 1 is not dealing with ROS production in photosynthesis, use a more appropriate reference.

line 62: the authors refer to the sak1 line that encodes a putative transcription factor involved in the acclimation response to 1O2. Is it possible to compare their transcriptomics data to the one of the sak1 mutant subjected to oxidative stress treatment?  Have the authors identified the sak1 transcript in the gpx5, control and complemented strain and how is the transcript evolving during the stress in these lines? 

Figure 1 D and E: have the authors checked cell size of the strains during the CdCl2 and HgCl2 treatment compared to C?

Lines 243-247: GOX transcripts encode glyoxal or galactose oxidase (phytozome), thus not a glycolate oxidase. In Chlamydomonas, a glycolate dehydrogenase catalyses the transformation of glycolate into glyoxylate (not a glycolate oxidase like in higher plants). What is  the 'glycolic acid cycle' mentioned by the authors? Reference 3 is not specific to Chlamydomonas. This paragraph should be revised in the light of the role of the GOX in Chlamydomonas. Chlororespiration is usually not considered as important in Chlamydomonas since there is an efficient CCM. By the way, have the authors detected modifications in the CCM, which could suggest a modification in the O2/CO2 levels and thus a shift towards chlororespiration? Have they found transcripts of the chlororespiration pathway modified?

Lines 246 and line 254: both sentences concern oxidative stress in gpx5 but it is not clear if the authors would like to say that gpx5 is more stressed or not.

line 270: MAS1 is one of the 2 enzymes specific of the glyoxylate cycle. This cycle is required for acetate assimilation. Have the authors also detected a variation in the amount of the ICL1 transcript? This would simply suggest that since gpx5 is growing more slowly than the control, the transcripts encoding transcripts involved in the glyoxylate cycle are in lower amounts. It would thus be also interesting to check CIS2 and ACS3, that are specific to the glyoxylate cycle.

Lines 276-283: it is tempting to extrapolate from the transcript amounts to the enzyme activity (lines 282-283) but in the absence of any lipid analysis, it is not advisable. The transcriptomics data only give indication on the relative amounts of transcripts. The authors should have this in mind everywhere.

Lines 291-295: isn't it expected that the transcripts of the cell cycle are in lower amounts since gpx5 is dividing less than the control?

Have the authors checked that the complemented strain (L27) entire transcriptome gave similar response to WT?

Lines 393-401: Have the authors checked the entire pathway of Jasmonate synthesis? Could the authors provide evidence about Jasmonate detection in Chlamydomonas? It seems controversial (see Han JEB 2017 Evolution of jasmonate biosynthesis and signaling mechanisms).

Lines 483-491: it may be interesting to measure whole cell respiration to make a link between the transcriptomics data and physiology. If this is true, it could also be a compensatory mechanism to counteract the decrease in photosynthetic activity, specifically in the presence of acetate.

A figure summarising when and in which conditions (+/- RB) the transcriptomics analyses  have been made in WT, complemented and gpx5 mutant would help the reader.

Fig 7: in the absence of any measurement/detection of glycolate, it is not advisable to write the conclusions like that and to make a link between glycolate and peroxisome. Is there any APX putatively localised in peroxisome (check for PTS1 or PTS2)? Is APX1 localised in the cytosol? Have the authors checked for the putative localisation (using PredAlgo, for example). 

Legend: AOX is mentioned but is not found on the figure. Concerning SRX; the authors mentioned in the text SRX1 which is not localised in the cytosol but putatively in the chloroplast (PredAlgo website). Please see the role of srx1 in plants.

Author Response

Comments and Suggestions for Authors

The paper contains a considerable amount of work, mainly transcriptomics data, aiming at characterising the gpx5 mutant of Chlamydomonas reinhardtii following oxidative stress. The paper requires improvements.

 Thanks for reviewer`s valuable comments and suggestions. We added supplementary figure S2 to show the experimental design and analysis, figure S6 to demonstrate the high similar genome-wide expression pattern between CC4348 and the complementary strain L27 after oxidative stress for 30 min. We analysed the chlororespiration, glyoxylate cycle and pathway of jasmonic acid synthesis.

 Point 1: line 39: reference 1 is not dealing with ROS production in photosynthesis, use a more appropriate reference.

Response 1:

We added reference 2 and 3 to introduce ROS production in photosynthesis.

Revised as follows:

  1. Del Rio, L. A., ROS and RNS in plant physiology: an overview. Journal of experimental botany 2015, 66 (10), 2827-37.
  2. Foyer, C. H., Reactive oxygen species, oxidative signaling and the regulation of photosynthesis. Environmental and experimental botany 2018, 154, 134-142.

Point 2: line 62: the authors refer to the sak1 line that encodes a putative transcription factor involved in the acclimation response to 1O2. Is it possible to compare their transcriptomics data to the one of the sak1 mutant subjected to oxidative stress treatment? Have the authors identified the sak1 transcript in the gpx5, control and complemented strain and how is the transcript evolving during the stress in these lines?

Response 2:

(1) Yes, we compared the transcriptomics data from the sak1 mutant and the gpx5 mutant subjected to oxidative stress treatment using 1 µM RB for 1 h. These analyses were shown in the discussion part 4.5. (new lines 820-831)

‘Compared the transcriptomics data from the gpx5 mutant and the sak1 mutant subjected to oxidative stress treatment using 1 µM RB for 1 h, there were 456 overlap DEGs which were induced in both the gpx5 mutant and the sak1 mutant, while 106 overlap DEGs depressed.’

(2) Yes, we identified the SAK1 transcript in the gpx5, parental and complemented strains.

After treatment with 1 µM RB for 1 h, the SAK1 transcript increased in abundance from 6 RPKM to 36 RPKM in CC4348, from 6 RPKM to 107 RPKM in the gpx5 mutant, and from 14 RPKM to 179 RPKM in complemented strain L27. Besides, we also found the GPX5 transcript in the sak1 mutant from the published data, which increased in abundance from 314 RPKM to 1112 RPKM. These data indicated that SAK1 protein and GPX5 protein function in the different pathway to respond to oxidative stress. (new lines 832-837)

Point 3: Figure 1 D and E: have the authors checked cell size of the strains during the CdCl2 and HgCl2 treatment compared to C?

Response 3:

According to the comments of reviewer 3, these results is not quite relevant to main topic of this manuscript, we removed the parts of figure 1D and 1E.

Point 4: Lines 243-247: GOX transcripts encode glyoxal or galactose oxidase (phytozome), thus not a glycolate oxidase. In Chlamydomonas, a glycolate dehydrogenase catalyses the transformation of glycolate into glyoxylate (not a glycolate oxidase like in higher plants). What is the 'glycolic acid cycle' mentioned by the authors? Reference 3 is not specific to Chlamydomonas. This paragraph should be revised in the light of the role of the GOX in Chlamydomonas. Chlororespiration is usually not considered as important in Chlamydomonas since there is an efficient CCM. By the way, have the authors detected modifications in the CCM, which could suggest a modification in the O2/CO2 levels and thus a shift towards chlororespiration? Have they found transcripts of the chlororespiration pathway modified?

Response 4:

(1) We adapted 'glycolic acid cycle' from the higher plant. The reviewer is right that the glycolate dehydrogenase catalyses functions in the transformation of glycolate into glyoxylate in Chlamydomonas, (not a glycolate oxidase like in higher plants).

(2) Yes, we found modifications in the CCM occurred in the mutant. Most of genes encoding proteins involving in CCM were up-regulated in the gpx5 mutant, including some low-CO2-inducible proteins (CCP2, LCI24, LCI5, LCIB and LCIE).

(3) Yes, we also re-analysed the expression change of genes in the chlororespiraton. Among them, 7 out of 9 transcripts showed no difference in abundance between CC4348 and the gpx5 mutant, including glycolate dehydrogenase (GYD1, Cre06.g288700), phosphoglycolate phosphatase (PGP1 and PGP2 encoding by Cre03.g168700 and Cre10.g438100, respectively), serine hydroxymethyltransferase (SHMT1, Cre16.g664550), alanine-glyoxylate transaminase (AGT1 and AGT2) and serine glyoxylate aminotransferase (SGA1). Only hydroxypyruvate reductase (HPR1) and alanine aminotransferase (AAT1) were down-regulated in the gpx5 mutant. These results meant the deficiency of GPX5 had few effects on chlororespiraton.

Point 5: Lines 246 and line 254: both sentences concern oxidative stress in gpx5 but it is not clear if the authors would like to say that gpx5 is more stressed or not.

Response 5:

When cells were grown on TAP complete medium under our optimized laboratory growth conditions,the transcript abundances of GSTs remained relatively low, such as GST1 and GST10 (4 and 6 RPKM, respectively). As the biomarkers of the response to stress, the expression of GSTs wasn`t induced compared to the parental strain CC4348. So we predicted that the gpx5 mutant didn`t suffer the stress conditions.

Point 6: line 270: MAS1 is one of the 2 enzymes specific of the glyoxylate cycle. This cycle is required for acetate assimilation. Have the authors also detected a variation in the amount of the ICL1 transcript? This would simply suggest that since gpx5 is growing more slowly than the control, the transcripts encoding transcripts involved in the glyoxylate cycle are in lower amounts. It would thus be also interesting to check CIS2 and ACS3, that are specific to the glyoxylate cycle.

Response 6:

Yes, we have also detected the ICL1 (isocitrate lyase) transcript, which decreased in abundance from 785RPKM in CC4348 to 282 RPKM in the gpx5 mutant under laboratory growth conditions. So were the transcripts of CIS2 (citrate synthase) and ACS3 (acetyl-CoA synthetase/ligase). The transcript of CIS2 and ACS3 decreased from 80 RPKM to 62 RPKM, and from 2508 RPKM to 1361 RPKM, respectively. These data would simply suggest that since gpx5 is growing more slowly than the control, the transcripts encoding transcripts involved in the glyoxylate cycle are in lower amounts.

We added the above paragraph in the text (new lines 317-323).

Point 7: Lines 276-283: Lack of GPX5 also affected lipid metabolism, resulting in 27 lipid-related DEGs (9 up-regulated and 18 down-regulated). The transcript abundance of KAS2 and KAS3, which encode beta ketoacyl acyl-carrier-protein synthase II and synthase III, respectively, and HAD1, encoding 3-hydroxyacyl- carrier-protein dehydratase, decreased to 0.07- to 0.35-fold in the gpx5 mutant. The transcripts of FAD6, encoding chloroplast omega-6-fatty acid desaturase, increased in abundance by 2.3-fold, reaching 1374 RPKM. Increasing the unsaturation of membrane lipids in cyanobacteria can improve the tolerance to low temperature. Therefore, increasing the fatty acid unsaturation by up-regulation of fatty acid desaturase may help the cell to tolerate oxidative stress in the gpx5 mutant.

it is tempting to extrapolate from the transcript amounts to the enzyme activity (lines 282-283) but in the absence of any lipid analysis, it is not advisable. The transcriptomics data only give indication on the relative amounts of transcripts. The authors should have this in mind everywhere.

Response 7:

Revised as follows in new lines 335-337:

Therefore, up-regulation the expression of genes encoding fatty acid desaturases may affect the metabolism of unsaturated fatty acid in the gpx5 mutant.

Point 8: Lines 291-295: isn't it expected that the transcripts of the cell cycle are in lower amounts since gpx5 is dividing less than the control?

Response 8:

Yes, the transcripts of genes related to the cell cycle and structural proteins were significantly down-regulated in the gpx5 mutant (Figure 2C), which was consistent with the slower division and growth.

Point 9: Have the authors checked that the complemented strain (L27) entire transcriptome gave similar response to WT?

Response 9:

We added a supplementary figure S6 to show the expression pattern of the parental strain CC4348 and the complementary strain L27 after treatment with 1 μM RB for 30 min. Among 322 up-regulated DEGs in CC4348 after treatment for 30 min, roughly 79% (253) overlapped with induced DEGs in L27. Among 386 down-regulated DEGs in CC4348 after treatment for 30 min, roughly 88% (341) overlapped with depressed DEGs in L27. Overall, about 84% (594 out of 708) DEGs in CC4348 were overlapped with that in L27 after treatment for 30 min.

Point 10: Lines 393-401: Have the authors checked the entire pathway of Jasmonate synthesis? Could the authors provide evidence about Jasmonate detection in Chlamydomonas? It seems controversial (see Han JEB 2017 Evolution of jasmonate biosynthesis and signaling mechanisms).

Response 10:

We have checked the homologous genes involved in pathway of jasmonate synthesis. There were seven predicted enzymes involved in jasmonic acid biosynthesis induced after RB treatment. The similarity between 13S-lipoxygenase (13-LOX, Cre12.g512300) in C. reinhardtii to lipoxygenase (LOX1, AT1G55020) in Arabidopsis thaliana, 12-oxophytodienoic acid reductase (OPR, Cre03.g210513) in C. reinhardtii to oxophytodienoate-reductase 3 (OPR3, AT2G06050) in Arabidopsis thaliana is 36% and 46%, respectively, so we are convinced to conclude that jasmonic acid biosynthesis occurred in C. reinhardtii.

Point 11: Lines 483-491: it may be interesting to measure whole cell respiration to make a link between the transcriptomics data and physiology. If this is true, it could also be a compensatory mechanism to counteract the decrease in photosynthetic activity, specifically in the presence of acetate.

Response 11:

Good suggestion, we found a type II NAD(P)H dehydrogenase (NDA1), an alternative oxidase (AOX1), complex IV (SCO1 and PET191) up-regulated, which fits well the data published by Sabeeha S. Merchant. They showed transcripts encoding proteins of the mitochondrial ETC genes up-regulated and the O2 production decreased.

(Blaby, I. K.; Blaby-Haas, C. E.; Perez-Perez, M. E., et al., Genome-wide analysis on Chlamydomonas reinhardtii reveals the impact of hydrogen peroxide on protein stress responses and overlap with other stress transcriptomes. The Plant journal: for cell and molecular biology 2015, 84 (5), 974-988.)

Point 12: A figure summarising when and in which conditions (+/- RB) the transcriptomics analyses have been made in WT, complemented and gpx5 mutant would help the reader.

Response 12:

According to the review`s suggestion, the diagram for experimental design and analysis was shown in supplementary figure S2 in the text.

CC448 and the gpx5 mutant cells were grown to a density of 2 x 106 cells mL-1 at 25°C under constant illumination of 60 µmol×m-2×s-1 with continuous shaking at 120 rpm. CC4348 and the gpx5 mutant cells were harvested for 0` time point and then Rose Bengal (RB) was added to a final concentration of 1 µM. Subsequent samples were taken at 30 min after incubation with RB. The cells were kept in continuous shaking at 120 rpm at 25°C under constant illumination of 60 µmol×m-2×s-1.

(A) The transcriptomes of CC448 and the gpx5 mutant from samples of 0` timepoint were analysed at first, regarded as samples under laboratory growth conditions, and these results were shown in figure 2.

(B) The transcriptomes of CC448 cells at 0` and 30 min after treatment were compared, as shown in figure 4.

(C) The different transcriptomes of CC448 and the gpx5 mutant after treatment for 30 min were compared, as shown in figure 5.

Point 13: Fig 7: in the absence of any measurement/detection of glycolate, it is not advisable to write the conclusions like that and to make a link between glycolate and peroxisome. Is there any APX putatively localized in peroxisome (check for PTS1 or PTS2)? Is APX1 localized in the cytosol? Have the authors checked for the putative localization (using PredAlgo, for example).

Response 13:

There are three APXs, including APX1 (Ascorbate peroxidase, Cre02.g087700), APX2 (L-ascorbate peroxidase, Cre06.g285150), APX4 (L-ascorbate peroxidase/Ascorbic acid peroxidase, Cre05.g233900). The APX1 localized in cytoplasmic using TargetP software.

Point 14: Legend: AOX is mentioned but is not found on the figure. Concerning SRX; the authors mentioned in the text SRX1 which is not localised in the cytosol but putatively in the chloroplast (PredAlgo website). Please see the role of srx1 in plants.

Response 14:

AOX was removed in the figure legends.

Sulfiredoxin (SRX1) is primarily located in the cytosol and it gets translocated to the mitochondria during increased oxidative burden. Sulfiredoxin is a member of the oxidoreductases family which plays a crucial role in thiol homoeostasis when under oxidative stress.

Reviewer 3 Report

The manuscript „Transcriptomic and physiological responses to oxidative stress in a Chlamydomonas reinhardtii glutathione peroxidase mutant“ compares the reaction to singlet oxygen treatment imposed by application by Rose Bengal on the physiology of Chlamydomonas reinhardtii. The study brings insights into cell´s response to physiologically occurring reactive oxygen species and is of interest to the fields of stress response as well as generally to algal physiology and biotechnology. The manuscript is overall well written and easy to follow. However, currently it suffers numerous minor errors such as 1) in places, it lacks sufficient details in the description of the experiments, 2) the conclusions derived are not sufficiently supported by the data. More specific comments are outlined in the PDF version of the manuscript. Furthermore, there are a few suggestions for the organization of the MS.

  • It is not clear what is the significance of the experiments with cadmium and mercury treatments? The experiments are presented in a single graph each and seemingly have no connection to the rest of the MS and are not discussed. I would suggest to remove them.
  • It is not clear why completely different algal strain was used for measuring flagella length? The strain is not related to the parental strain so the significance of its use is unclear and it is not discussed. The relevance of the results is questionable even if there would be some significance. Moreover, from the presented data, it seems there is no connection between flagella length and reaction to oxidative stress. Since I understand the limitations of measuring flagella length in CC4348 and its derivatives, I would suggest using different type of assay. For example, it would be informative to present some data on cell numbers and possibly cell size. Both of which are very easy assays that can be based on light microscopy only. This might strengthen the conclusions derived in the paper.
  • In my understanding, the main message of the MS is to pinpoint differences in genes expressed in parental strain and in gpx5 mutant. If so, I would suggest adding some kind of summary or conclusion, which would state the similarities and differences between the two in terms of differential gene expression. For instance, from the data it is clear that alternative pathways are employed in gpx5 mutant upon the Rose Bengal treatment but this is never explicitly stated. It would be helpful to have some of the differently expressed genes pinpointed together with suggestions on how this might be significant for the cell´s response.

Author Response

Comments and Suggestions for Authors

The manuscript ‘Transcriptomic and physiological responses to oxidative stress in a Chlamydomonas reinhardtii glutathione peroxidase mutant’ compares the reaction to singlet oxygen treatment imposed by application by Rose Bengal on the physiology of Chlamydomonas reinhardtii. The study brings insights into cell´s response to physiologically occurring reactive oxygen species and is of interest to the fields of stress response as well as generally to algal physiology and biotechnology. The manuscript is overall well written and easy to follow. However, currently it suffers numerous minor errors such as 1) in places, it lacks sufficient details in the description of the experiments, 2) the conclusions derived are not sufficiently supported by the data. More specific comments are outlined in the PDF version of the manuscript. Furthermore, there are a few suggestions for the organization of the MS.

Thanks for reviewer`s valuable comments and suggestions. We focused on adding sufficient details of methods, such as isolation of total RNA and pooling sequencing libraries. Moreover, to state explicitly the similarities and differences responses to oxidative stress between CC43438 and the gpx5 mutant, we added a brief summary following the results and simplify the model figures.

 Point 1: It is not clear what is the significance of the experiments with cadmium and mercury treatments? The experiments are presented in a single graph each and seemingly have no connection to the rest of the MS and are not discussed. I would suggest to remove them.

Response 1:

Agree, the whole paragraph of the experiments with cadmium and mercury treatments was removed.

Point 2: It is not clear why completely different algal strain was used for measuring flagella length? The strain is not related to the parental strain so the significance of its use is unclear and it is not discussed. The relevance of the results is questionable even if there would be some significance. Moreover, from the presented data, it seems there is no connection between flagella length and reaction to oxidative stress. Since I understand the limitations of measuring flagella length in CC4348 and its derivatives, I would suggest using different type of assay. For example, it would be informative to present some data on cell numbers and possibly cell size. Both of which are very easy assays that can be based on light microscopy only. This might strengthen the conclusions derived in the paper.

Response 2:

(1) CC4348 (sta6-1 mt+) strain is the cell-wall-deficient, flagellum-free, and starchless mutant lacking the small subunit of the heterotetrameric ADP-glucose pyrophosphorylase (https://www.chlamycollection.org/). Because of high oil accumulation under N-starvation, CC4348 was initially used to screen the mutants with defects in lipid droplets formation, the gpx5 mutant was one of them. Because CC4348 and its derivative gpx5 mutant lack cell wall and are difficult to be crossed to obtain the cell with flagella. So, in our study, widely used wild type CC125 strain was employed for measuring flagella length.

(2) In the context, we found the genes encoding flagellar associated proteins were down-regulated after oxidative stress both in CC4348 and the gpx5 mutant. We wondered if oxidative stress indeed affected the expression of flagellar associated proteins. To confirm them, the number and length of flagellar were measured. The length of flagella shown little effects to oxidative stress may due to big pool of mRNA of flagellar protein exits. Since the rate of flagella cells was decreased, we speculated oxidative stress affected the deflagellation.

(3) We agreed the detection of cell numbers and cell size would be another way to testify the effect of oxidative stress. In our previous experiment, the size of CC4348 and the gpx5 mutant cells shown no difference under nitrogen difference. (Miao, R.; Ma, X.; Deng, X., et al., High level of reactive oxygen species inhibits triacylglycerols accumulation in Chlamydomonas reinhardtii. Algal Research 2019, 38, 101400.). We also found the good correlation between the cell number and OD750nm as shown in supplementary figure S1 in the text.

Point 3: In my understanding, the main message of the MS is to pinpoint differences in genes expressed in parental strain and in gpx5 mutant. If so, I would suggest adding some kind of summary or conclusion, which would state the similarities and differences between the two in terms of differential gene expression. For instance, from the data it is clear that alternative pathways are employed in gpx5 mutant upon the Rose Bengal treatment but this is never explicitly stated. It would be helpful to have some of the differently expressed genes pinpointed together with suggestions on how this might be significant for the cell´s response.

Response 3:

Revised as follows in new lines 635-641:

‘In brief, after RB treatment, CC4348 and the gpx5 mutant shared some common pathways to respond to oxidative stress, including up-regulated ROS detoxification, ubiquitin mediated proteolysis (subgroup A1), and inhibited the expression of cell structure related proteins and selenoproteins (subgroup C3). When GPX5 protein was deficiency, some specific responses occurred, such as accelerated the TCA cycle and mETC in mitochondria to supply more ATP, repressed genes related to chlorophyll metabolism and photosynthesis (subgroup A2 and A3), and down-regulated iron transporters (subgroup B3).’

Point 4: Lines 75-78:All cells were cultured mixotrophically in Tris-acetate-phosphate (TAP) medium (20 mM Tris, 0.4 mM MgSO4×7H2O, 0.34 mM CaCl2×2H2O, 10 mM NH4Cl, 1 mM phosphate, 10 mL/L glacial acetic acid and trace elements) at 25°C under constant illumination of 60 µmol×m-2×s-1 with continuous shaking at 120 rpm. Cells were grown on solidified TAP medium containing 1.5% agar.

(1) What was the initial and final cell concentration?

(2) Was this incident light?

(3) Please specify the conditions, temperature, light intensity, duration.

Response4:

(1) All cells were cultured from the initial concentration at 1.7 x 106 cells×mL-1 (OD750 nm = 0.5). The final cell concentration varied from different cells, but almost all for 4 days. The OD750nm of CC4348 reached about 1.6, while the gpx5 mutant 1.2 and the complemented strain L27 1.4.

(2) Yes, it is incident light.

(3) Cells were grown on solidified TAP medium containing 1.5% agar at 25°C under constant illumination of 60 µmol×m-2×s-1 for about 5 days.

Revised as follows:

As the initial concentration at 1.7 x 106 cells×mL-1, all cells were cultured mixotrophically in Tris-acetate-phosphate (TAP) medium (20 mM Tris, 0.4 mM MgSO4×7H2O, 0.34 mM CaCl2×2H2O, 10 mM NH4Cl, 1 mM phosphate, 10 mL/L glacial acetic acid and trace elements) at 25°C under constant incident illumination of 60 µmol×m-2×s-1 with continuous shaking at 120 rpm. Cells were grown on solidified TAP medium containing 1.5% agar at 25°C under constant incident illumination of 60 µmol×m-2×s-1 for about 5 days.

Point 5: Lines 80-85: For the RB treatments, 100 mL of liquid culture was harvested at the mid-exponential phase (about OD750 = 0.6) by centrifugation at 2500 g for 3 min at room temperature. The supernatant was discarded, the cells were washed in fresh TAP medium one time, and resuspended in an equal volume of fresh culture medium, and fresh RB (Sigma Aldrich) was added to a final concentration of 1 µM. Samples taken immediately upon inoculation with RB-containing media were labeled as 0 min and subsequent samples were taken after 15, 30, 45, and 60 min incubation and labeled accordingly.

(1) Approximately number of cells?

(2) What was the sample size? How were the cultures maintained in between the sampling?

Response 5:

(1) The number of cells is about 2 x 106 cells mL-1 when OD750nm reaches at 0.6.

(2) In the experiment of RB treatment, we didn`t measure the sample size. But in our previous experiment of nitrogen deprivation, the size of CC4348, the gpx5 mutant cells and other complemented strains (C14, C19, C108, C203, L5, S33 and TP128) shown no difference. (Miao, R.; Ma, X.; Deng, X., et al., High level of reactive oxygen species inhibits triacylglycerols accumulation in Chlamydomonas reinhardtii. Algal Research 2019, 38, 101400.) So we predicted there were same number of cells when culture had the same OD750 value.

Point 6: Lines 91-92: Samples were taken for measurement of OD750nm every 12 hours.

How were the samples maintained during the experiment?

Response 6:

The paragraph described the experiment of heavy metal treatment including lines 91-92 were removed.

Point 7: Lines 94-97:Cells were stained with 2′,7′-dichlorofluorescein diacetate (DCFH-DA) (Sigma Aldrich) as described in our previous study 14, The nonpolar, nonionic DCFH-DA that becomes nonfluorescent DCFH after crossing cell membranes and flowing enzymatically hydrolyzed by intracellular esterases. Hydrogen peroxide oxidized DCFH to fluorescent dichlorofluorescein (DCF).

(1) following?

(2) How was the fluorescence measured?

Response 7:

(1) The word ‘following’ was revised to ‘being’.

(2) The fluorescence of DCF was detected using multi-mode microplate reader (Filter Max F5, Molecular Devices; excitation at 485 nm and emission at 530 nm).

Point 8: Lines 103-104: After all samples were collected, RNA was prepared by thawing (10 min at 24°C) following the manufacturer’s instructions.

Please add some more details here.

Response 8:

After all samples were collected, the sample resuspended in Trizol were taken out, thawed at 24°C for about 10 min. In proportion to the volume of Trizol, 1 mL chloroform was added to the suspension, rocked for 15 s following by maintaining on ice for 3 min. After centrifuged at 12000 g for 15 min at 4 °C, the supernatant was saved to add 2.5 ml isopropanol following by maintaining on ice for 10 min. After centrifuged at 12000 g for 10 min at 4 °C, the supernatant was washed by 75% ethanol, dried, and dissolved in diethyl pyrocarbonate-treated double-distilled H2O (ddH2O).

Point 9: Lines 105-107: The RNA concentration was measured on a NanoDrop 2000 spectrophotometer (Thermo Scientific), and the quality of the RNA was assessed on an Agilent 2100 Bioanalyzer.

How were the sequencing libraries prepared? What type of RNA was analysed?

Response 9:

To analyse the gene expression of the whole-genome, RNA samples from CC4348, the gpx5 mutant and the complemented strain L27 were isolated and used to construct the sequencing libraries for Illumina sequencing. Briefly, mRNA was purified from 5 μg of total RNA using oligo (dT) magnetic beads, broken into 450-550 bp fragments, and added unique adapter for every sample using the TruSeq RNA sample preparation kit (Illumina) according to the manufacturer’s instructions.

Point 10: Lines 109-110: RNAs were sequenced at Illumina for estimating the transcript abundance in the RB incubation time-course experiments.

How was the sequencing done? Length of sequenced, single end, pair end?

Response 10:

The pooled sequencing libraries were sequenced on the Illumina HiSeq X-ten platform for 150 bp pair end sequencing mode. About 10 G sequencing data were obtained per library for each sample.

Point 11: Lines 144-145: After treatment with 1 μM RB for 30 minutes, aliquots of CC125 cells were fixed with 0.5% Lugol’s solution.

Is this strain name correct? It doesnt fit with the above.

Response 11:

Yes, the strain name ‘CC125’ is correct.

CC4348 (sta6-1 mt+) strain is the cell-wall-deficient, flagellum-free, and starchless mutant lacking the small subunit of the heterotetrameric ADP-glucose pyrophosphorylase (https://www.chlamycollection.org/). Because of high oil accumulation under N-starvation, CC4348 was initially used to screen the mutants with defects in lipid droplets formation, the gpx5 mutant was one of them. BecauseCC4348 and its derivative gpx5 mutant lack cell wall and are difficult to be crossed to obtain the cell with flagella. So, in our study, widely used wild type CC125 strain was employed for measuring flagella length.

Point 12: Lines 160-162: In the complemented strain L27, the level of ROS increased to 2.5-fold, which was intermediate between the parental strain and the mutant (Figure 1A). These results demonstrated that GPX5 plays an important role in the removal of ROS.

How was the normalization done? Is all normalized to the zero hour of the untreated control cells? Or was the normalization done for each strain independently?

Response 12:

Under oxidative stress condition, the level of ROS was normalized for each strain from untreated condition independently.

Point 13: Lines 169-173: Figure 1 D: Comparison of the growth of CC4348, the gpx5 mutant, and L27 cells treated with 500 μM CdCl2 versus without treatment. The error bars represent standard deviations of three biological replicates. E: Comparison of the growth of CC4348, the gpx5 mutant, and L27 cells treated with 12 μM HgCl2 versus without treatment. The error bars represent standard deviations of three biological replicates.

Which of the curves is without treatment?

Response 13:

According to the reviewer`s suggestion, the experiment and results (figure 1D and 1E) were removed.

Point 14: Lines 178-181: The spots of the CC4348 strain were much larger and greener after 4 days, compared with the other strains. Indeed, the gpx5 mutant grew more slowly after being treated with 2 μM RB and was unable to survive 4 μM RB treatment. The two complemented strains, L27 and C203, grew similarly to CC4348.

(1) This is not true, the parental strain and the complemented strain C203 look quite comparably even at the highest concentration.

(2) This cannot be judged from the figure, there the mutant and the complemented strain L27 look the same.

(3) This is in clear contradiction to what is said above.

Response 14:

(1) To be precisely, the spot of the CC4348 strain were much larger and greener after 4 days compared with the gpx5 mutant.

(2) The colour of the spot of the mutant is slightly lighter than that of the complemented strain, and more obvious lighter than that of the parental strain.

(3) The spot of the CC4348 strain were much larger and greener after 4 days compared with the gpx5 mutant, not with the two complemented strains. The two complemented strains grew similarly to CC4348.

Revised as follows:

The spots of the CC4348 strain were much larger and greener after 4 days, compared with the gpx5 mutant. Indeed, the gpx5 mutant grew more slowly after being treated with 2 μM RB compared to the CC4348 and was unable to survive 4 μM RB treatment. The two complemented strains, L27 and C203, grew similarly to CC4348.

Point 15: Lines 185-189: CC4348 cells reached a stable period after 72 h of incubation, and the OD750 reached 1.6. However, the growth of gpx5 cells was compromised and the OD750 was only 1.2 when it reached a stable stage. The growth status of L27 was in between that of CC4348 and gpx5, and the OD750 was 1.4 when it reached a stable stage (Figure 1C). These data indicated that the deficiency of GPX5 effected cell cycle and lowered the population density.

OD is only indirect measure, what was the cell density in fact?

Response 15:

By comparing the OD750nm and cell number, we found that these two parameters had the good linear correlation in a C. reinhardtii culture. Please see the following figure of the relation between the OD750nm values and the cell numbers.

Point 16: Lines 194-196: However, the gpx5 mutant cells stopped growing after 36 h and the maximum OD750 was only 0.8, which is much lower than that of CC4348.

Again, it would be useful to have cell counts and possibly cell sizes to support these statements. As of now they are very indirect. To make a fair comparison, growth rates or doubling times should be calculated from the exponential phase. The same applies to the effect of mercury.

Response 16:

By comparing the OD750nm and cell number, we found that these two parameters had the good linear correlation in a C. reinhardtii culture. Besides, the size between CC4348, the gpx5 mutant and complemented strains has no difference. Moreover, in our previous experiment of nitrogen deprivation, the size of CC4348, the gpx5 mutant cells and other complemented strains (C14, C19, C108, C203, L5, S33 and TP128) shown no difference. (Miao, R.; Ma, X.; Deng, X., et al., High level of reactive oxygen species inhibits triacylglycerols accumulation in Chlamydomonas reinhardtii. Algal Research 2019, 38, 101400.) So we predicted the gpx5 mutant with lower OD750nm value would have the less cell number compared to that of CC4348.

Point 17: Lines 202-205: To assess the transcriptome of the gpx5 mutant under optimized laboratory growth conditions (no extra oxidative stress added), we performed RNA-seq on CC4348 and gpx5 cells grown in TAP complete medium at 25°C under constant illumination of 60 µmol×m-2×s-1 with continuous shaking at 120 rpm.

When was the sampling done? In exponential or stationary phase? Please clarify.

Response 17:

(1) All cells were grown at a density of 2 x 106 cells mL-1in exponential phase before collecting as an initial sample for RNA-sequence (0`).

Point 18: Lines 230-232: Figure 2 D: The maximum efficiency of PSII (Fv/Fm) of CC4348, the gpx5 mutant, and L27 cells under laboratory growth conditions.

At what timepoint was the ratio measured?

Response 18:

The maximum efficiency of PSII (Fv/Fm) under laboratory growth conditions was measured when cell grown at a density of 2 x 106 cells mL-1in exponential phase.

Point 19: Lines 248-249: Transcript abundances of the glutathione S-transferase (GST) genes (GST1 and GST10) were very low in the gpx5 mutant (4 and 6 RPKM, respectively).

Compare to what RPKM in parental strain?

Response 19:

Revised as follows in the text:

Transcript abundances of GST1 and GST10 in CC4348 under laboratory growth conditions were 19 RPKM and 32 RPKM, respectively. Either the transcript abundance of GST1 (4 RPKM) or GST10 (6 RPKM) was low in the gpx5 mutant than that in CC4348.

Point 20: Lines 264-266: The results showed a reduction in photosynthetic efficiency of the gpx5 mutant compared to CC4348, suggesting that the gpx5 mutant might minimize ROS accumulation by slowing down photosynthesis, which is a major source of ROS.

Is this statistically significant?

Response 20:

No, this is not statistically significant. The Fv/Fm value of CC4348 under laboratory growth conditions was 0.68, while 0.63 in the mutant. It`s difference was too little to be significantly.

Revised as follows:

The results showed a slightly reduction in photosynthetic efficiency of the gpx5 mutant compared to CC4348, suggesting that the gpx5 mutant might minimize ROS accumulation by slowing down photosynthesis, which is a major source of ROS.

Point 21: Lines 292-295: These effect on cell cycle-related genes is consistent with the effect on growth described above (Figure 1C). These results suggested that the deficiency of GPX5 in the mutant not only affected ROS homeostasis, but also affected cell growth.

(1) The effects on cell cycle need to be documented better to be relevant.

(2) Which results? Does this fit the growth curves presented above? is there less or smaller cells in the mutant?

Response 21:

(1) Most of cell cycle-related genes were down-regulated in the gpx5 mutant compared to the parental strain CC4348, which fits well our observation that the mutant cells grew slower, and the concentration of cells in stationary phase was lower (Figure 1C).

(2) These results refer to “the grow difference between the mutant gpx5 and parental strain”

(3) Yes, down-regulated transcripts of genes related to cell cycle was fit the growth curves. Based on our previous publication, the size of CC4348 and the gpx5 mutant cells shown no difference. (Miao, R.; Ma, X.; Deng, X., et al., High level of reactive oxygen species inhibits triacylglycerols accumulation in Chlamydomonas reinhardtii. Algal Research 2019, 38, 101400.)

Point 22: Lines 302-305: Carbon metabolism, lipid metabolism and synthesis of chlorophyll were depressed, leading to lower efficiency of photosynthesis in the gpx5 mutant than in CC4348. These overall multifaceted changes in transcription slowed the growth of the gpx5 mutant cells.

(1) ‘lower efficiency’, Is this based?

(2) This might be well true but it needs to be evidenced clearly.

Response 22:

Revised as follows:

The expression of genes related to carbon metabolism, lipid metabolism, synthesis of chlorophyll and cell cycles were depressed, and the efficiency of photosynthesis decreased in the gpx5 mutant than in CC4348. These overall multifaceted changes slowed the growth of the gpx5 mutant cells.

Point 23: Lines 306: 3.3. Transcriptome responses to 1O2 in wild-type CC4348 cells

These are not wild type cells; it is a cell-wall mutant which also harbors a sta6-1 mutation. Parental strain? Please change accordingly throughout the text.

Response 23:

CC4348 strain is the parental strain of the gpx5 mutant.

Revised the word ‘wild-type CC4348’ to ‘parental strain CC4348’ in the manuscript.

Point 24: Lines 331-333: Figure 3 legends: B. An overview of the quantification of the change of transcript abundance of DEGs in Figure 3A. C. An overview of the quantification of the transcript abundance of DEGs in Figure 3A.

Please specify the difference between B and C.

Response 24:

Figure 3B showed the fold change of transcript abundance of DEGs in Figure 3A, while figure 3C showed the relative transcript abundance of DEGs in Figure 3A. The X-axis of figure 3B indicated the logarithm based 2 of the ratio of transcript abundance after treated with 1 μM RB for 30 minutes to control, while figure 3C showed the number of transcriptional RPKM after treated with 1 μM RB in CC4348 for 30 minutes.

Revised as follows:

B: An overview of the fold change of the transcript abundance of DEGs in Figure 3A. The X-axis indicated the logarithm based 2 of the ratio of transcript abundance of genes treated with 1 μM RB for 30 minutes compared to the transcript abundance of genes without treatment.

C: An overview of the relative the transcript abundance of DEGs in Figure 3A. The X-axis showed the number of transcriptional RPKM treated with 1 μM RB in CC4348 for 30 minutes.

Point 25: Lines 355: Figure 4. Transcriptional changes of antioxidant-related genes in RB-treated cells.

Specify the strain.

Response 25:

Transcriptional changes of antioxidant-related genes in RB-treated cells in CC4348.

Point 26: Lines 364-365: transcript abundance of HSP70A and HSP90A increased to 726 and 901 RPKM after treatment with RB for 30 min, respectively (Figure 3C and Table S3).

How big increase this is comparatively?

Response 26:

The transcript abundance of HSP70A increased about 3-fold from 261 RPKM to 726 RPKM after treatment with RB for 30 min, while HSP90A increased more than 3-fold from 275 RPKM to 901 RPKM.

Point 27: Lines 386-389: The expression of 21 transcripts related to lipid metabolism was up-regulated and only two genes were down-regulated. Transcripts encoding cyclopropane fatty acid synthase (CFA1 and CFA2) increased in abundance by about 19-fold and 32-fold to 217 and 70 RPKM, respectively (Figure 3B and C).

From the figure it is impossible to tell to what is referred to.

Response 27:

Figure 3B and 3C shown the 475 differently expressed genes from 18 functional classes of CC4348 after treatment with RB for 30 min. Since the dots were too small to be seen clearly,the supplementary table S3 included all the information of DEGs.

Revised as follows:

The expression of 21 transcripts related to lipid metabolism was up-regulated and only two genes were down-regulated. Transcripts encoding cyclopropane fatty acid synthase (CFA1 and CFA2) increased in abundance by about 19-fold and 32-fold to 217 and 70 RPKM, respectively (Figure 3B and C, Table S3).

Point 28: Lines 404-407: To further explore whether oxidative stress affected the structure and function of flagella, we examined wild-type CC125 cells treated with 1 μM RB for 1 h. Up to 37% of these cells lost their flagella at 12 h. The length of flagella did not change (Figure S2), suggesting that RB induced deflagellation, not resorption.

(1) Why should be used this unrelated wild type strain particularly for this assay? What is the relevance for the rest of the study?

(2) What is a significance of this? The change in flagellated cells is occuring much later than the detected changes in metabolism. Why should they be connected and not just fortuitous.

Response 28:

(1) CC4348 (sta6-1 mt+) strain is the cell-wall-deficient, flagellum-free, and starchless mutant lacking the small subunit of the heterotetrameric ADP-glucose pyrophosphorylase (https://www.chlamycollection.org/). Because of high oil accumulation under N-starvation, CC4348 was initially used to screen the mutants with defects in lipid droplets formation, the gpx5 mutant was one of them. BecauseCC4348 and its derivative gpx5 mutant lack cell wall and are difficult to be crossed to obtain the cell with flagella. So, in our study, widely used wild type CC125 strain was employed for measuring flagella length.

(2) Incidentally, we found the expression of flagellar associated proteins were inhibited under oxidative stress. Moreover, the ratio of cells with flagella decreased after oxidative treatment. So we predicted the relationship between RB treatment induced deflagellation. This interesting observation need to be further explored.

Point 29: Lines 416-418: To compare the responses of CC4348 and the gpx5 mutant, we divided the 1852 DEGs into three categories. Group A were up-regulated during the oxidative stress in CC4348 and included 947 genes.

The grouping is not entirely clear. The genes were up/down regulated in parental strain but what was their behaviour in the mutant?

Response 29:

At first, total 708 DEGs was obtained from CC4348 and 1639 DEGs were obtained from gpx5 mutant after treatment with RB for 30 min, respectively. Combined these 708 DEGs and 1639 DEGs, 1852 nonredundant genes were obtained. Secondly, these 1852 DEGs were grouped into group A, group B and group C just according to their expression in CC4348. Genes from each of these three group were up regulated, or down regulated, or not significantly changed. Thirdly, each group including group A, B and C were furtherly divided into subgroup according the expression in gpx5 mutant. For example, group A were divided into subgroup A1, A2 and A3.

In briefly, Group A were up-regulated during the oxidative stress in CC4348. Among them, some are also up-regulated (subgroup A1), not significantly changed (subgroup A2) and down-regulated (subgroup A3) in the mutant.

Point 30: Lines 436-438: Table 1. Overview of metabolic pathways and cellular processes for which related transcripts accumulated or declined in CC4348 and the gpx5 mutant cells after treatment with 1 μM RB for 30 min. The numbers in brackets represent the counts of according subgroups.

(1) Please add more details in the table description. The significance of the text and groups is not clear.

(2) Low ROS in table text. What is the meaning of this?

Response 30:

(1) Table 1 shown the overview of metabolic pathways and cellular processes changed in each subgroup from figure 5.

Subgroup A2 showed up-regulated in CC4348 and not significantly changed in the gpx5 mutant, while subgroup A3 up-regulated in CC4348 and down-regulated in the gpx5 mutant. Therefore, subgroup A2 and A3 were specifically up-regulated in CC4348.

Subgroup A1 showed up-regulated both in CC4348 and gpx5 mutant, namely shared induced DEGs.

Subgroup B1 showed no significantly changed in CC4348 but up-regulated in gpx5mutant, therefore, subgroup B1 were specifically up-regulated in the gpx5 mutant.

Subgroup B3 showed no significantly changed in CC4348 but down-regulated in gpx5 mutant, therefore, subgroup B3 were specifically down-regulated in the gpx5mutant.

Subgroup C2 showed down-regulated in CC4348 and not significantly changed in the gpx5mutant, therefore, subgroup C2 were specifically down-regulated in CC4348.

Subgroup C3 showed down-regulated both in CC4348 and the gpx5 mutant, therefore, namely shared inhibited DEGs.

(2) Low ROS indicated the ROS level were relatively lower in the CC4348 compared to that in the gpx5 mutant after the same treatment, as shown in Figure 1 A.

Point 31: Lines 567-573: Figure 6 legends: The response of photosynthesis to 1O2. A-C: The expression pattern of DEGs related to chlorophyll metabolism in CC4348 (A), the gpx5 mutant (B) and L27 (C) determined by RNA-seq. D- E: The expression pattern of DEGs encoding light-harvesting proteins in CC4348 (D), the gpx5 mutant (E) and L27 (F) determined by RNA-seq. The initial RPKM was set as 1 to normalize the data. Genes and corresponding RPKM values used in this analysis can be found in Table S6. G-H: Comparison of the maximum efficiency (Fv/Fm) of photosystem II (G) and content of chlorophyll (a + b) (H) in CC4348, the gpx5 mutant and L27.

(1) Please, add some explanatory legend directly to the figure.

(2) For each strain separately or to one of the strains?

(3) G and H are switched.

Response 31:

(1) The transcriptional response of genes related to photosystem to 1O2. A-C: The expression pattern of DEGs encoding light-harvesting proteins in CC4348 (A), the gpx5 mutant (B) and L27 (C) after treated with 1 μM RB determined by RNA-seq. D- E: The expression pattern of DEGs related to chlorophyll metabolism in CC4348 (D), the gpx5 mutant (E) and L27 (F) after treated with 1 μM RB determined by RNA-seq.

(2) The initial abundances of DEGs related to chlorophyll metabolism and encoding light-harvesting proteins were used to normalize the expression abundances of genes treated for RB for 30, 60 minutes in each strain separately.

(3) revised: G-H: Comparison of the content of chlorophyll (a + b) (G) and maximum efficiency (Fv/Fm) of photosystem II (H) in CC4348, the gpx5 mutant and L27.

Point 32: Lines 578-580: The expression of chlorophyll metabolism-related genes in CC4348 showed a continuous trend of up- regulation within 60 min of oxidative stress, whereas those in the gpx5 mutant showed an upward trend in expression within the first 30 min and remained stable in the following 30 min.

Was the increase significant? From the graph it seems very minor, below the treshold of 2.

Response 32:

Yes, the expression of CHLG and ALAD1 were up-regulated significantly in the gpx5 mutant (Figure 6E, Tables S4). The content of chlorophyll was increased significantly within the first 30 min in the gpx5 mutant (Figure 6G). Maybe the CHLG and ALAD1 played the vital roles in chlorophyll biosynthesis. After 30 min for treatment in gpx5, A feedback mechanism may work and chlorophyll biosynthesis scaled back.

Point 33: Lines 583-584: In CC4348, CHLG expression was up-regulated during RB treatment and reached a peak of 45 RPKM after 60 min;

Which translates to what up-regulation?

Response 33:

The transcript abundance of CHLG was continuously increased from 12 RPKM before treatment and reached a peak of 45 RPKM after 60 min in CC4348.

Point 34: Lines 621-622: The different expression pattern of genes with or without GPX5 indicated that cells have specific responses to different levels of stress.

in algal strains with? Please re-phrase.

Response 34:

Revised as follows:

The different expression pattern of genes with or without GPX5 function in algal Chlamydomonas indicated that cells have specific responses to different levels of stress.

Point 35: Lines 742-748: We also detected numerous transcripts encoding cell structure proteins, including FAPs, which were down-regulated after RB treatment in CC4348 and the gpx5 mutant. However, some of them increased in abundance in the sak1 mutant after RB treatment or in CC4532 after H2O2 treatment. For example, the transcripts encoding alpha tubulin (TUA1 and TUA2) and beta tubulin (TUB1 and TUB2) were down-regulated in CC348 and gpx5 after RB treatment, but they showed no significant change and high abundances (> 500 RPKM). Other factors might affect cell structure proteins besides oxidative stress.

This is unclear, please, re-phrase.

Response 35:

Revised as follows:

We also detected numerous down-regulated transcripts in CC4348 and the gpx5 mutant after RB treatment, including cell structure proteins, some of which up-regulated in the sak1 mutant after RB treatment or in CC4532 after H2O2 treatment. For example, the transcripts encoding FAP129 (Cre03.g199500) were down-regulated in CC348 and gpx5 after RB treatment, while increased more than 3-fold in abundance in the sak1 mutant under oxidative stress. The transcripts encoding FAP126 (Cre12.g536100) decreased in abundance from 72 RPKM to 3 RPKM in the gpx5 mutant, whereas remained about 70 RPKM in the the sak1 mutant after RB treatment or in CC4532 after H2O2 treatment. These data indicated that the SAK1 protein and GPX5 protein didn`t function in the same pathway and the expression of cell structure proteins may be regulated separately.

Round 2

Reviewer 2 Report

The authors have introduced some modifications which greatly help; however, many imprecisions are still present and should be corrected.

Lines 40-42

 ‘Photosystem II releases oxygen with H2O, accompanied by the production of ROS, which include singlet oxygen, superoxide, hydrogen peroxide, and hydroxyl radicals.’

Sentence is not clear: PSII does not release both O2 and H2O. In addition, PSI also participates to the production of ROS.

Lines 76-77

The description of the sta6 mutant is not made in reference 15. Please use the correct original reference and additional references describing the sta6 mutant. For example, the sta6 mutant has been shown not only defective for the small subunit of the heterotetrameric ADP-glucose pyrophosphorylase, being also affected in one RBO gene (Plant Cell. 2013 Nov;25(11):4305-23. doi: 10.1105/tpc.113.117580. Epub 2013 Nov 26. Systems-level analysis of nitrogen starvation-induced modifications of carbon metabolism in a Chlamydomonas reinhardtii starchless mutant. Blaby IK1, Glaesener AG, Mettler T, Fitz-Gibbon ST, Gallaher SD, Liu B, Boyle NR, Kropat J, Stitt M, Johnson S, Benning C, Pellegrini M, Casero D, Merchant SS.).

Please also consider the paper which describes the oxidative stress in the sta6 mutant.

Sci Rep. 2019 Jul 8;9(1):9856. doi: 10.1038/s41598-019-46313-6.

Impairment of starch biosynthesis results in elevated oxidative stress and autophagy activity in Chlamydomonas reinhardtii.

Tran QG, Cho K, Park SB, Kim U, Lee YJ, Kim HS. 

The fact that the sta6 mutant is also affected in its stress response should be considered for the comparison with gpx5.

The parental strain CC4348 is not cell wall less in the Chlamy collection. Please check.

Line 289: ‘re-analyzed’? please use ‘analyze’

Line 460: identity instead of similarity?

Line 558-562: Blaby et al. 2015 (Plant J, mentioned by the authors ref 42) showed that both PS and respiratory activities are decreased in WT after H2O2 treatment. This suggests that ATP produced by respiration is in less amount in stress conditions. The authors should revise their conclusions (These results suggested that the high concentration of ROS stimulates the mETC to provide more ATP – line 561-562) if they would like to rely on the results of Blaby et al or make respiratory measurements to clarify the point.

Line 561: consumption instead of production? Decreased and not 'decreasd'

As previously mentioned, the main enzyme catalyzing the conversion of P-glycolate into glyoxylate in Chlamydomonas is a glycolate dehydrogenase putatively localized in mitochondria (see Enzymatic characterization of Chlamydomonas reinhardtii glycolate dehydrogenase and its nearest proteobacterial homologue, Mohamed H. Aboelmy, Christoph Peterhansel* Plant Physiology and Biochemistry 79 (2014) 25e30). Please adapt the figures accordingly. What is the FPKM levels of the glycolate dehydrogenase in the different strains and conditions?

Line 836: ref 42 is not dealing with sak1.

Figure S2: poor quality, remove the background

Specific responses of the authors

Response 13

APX1

According to Urzicka et al 2012 JBC, APX1 would be located in CPL for removal of H2O2 from PSI in Chlamydomonas. According to Wu and Wang (2019) PLoS ONE 14(12): e0226543. https://doi.org/10.1371/journal. pone.0226543) APX1 would be located to CPL or mitochondria. The localization of APX1 is thus far to be clear. Please discuss and adapt figures according to the possibility that APX1 is not cytosolic.

Response 14:

SRX1

In Phytozome, 2 genes (Cre05.g232800 and Cre17.g729950) are named SRX1. SRX1 encoded by Cre17.g729950 is putatively localized in the CPL. Indeed, the homologue described by Phytozome in Arabidopsis is AT1G31170.1 which has been demonstrated to be chloroplastic (Rey et al Plant J 2007).

In addition, TargetP, mentioned above by the authors, predicts a chloroplast localization for SRX1 (Cre17.g729950) and the paper previously mentioned by the authors (Blably et al 2015 Plant J) also proposed a CPL localization of SRX1.

The homologue for the other SRX1 (Cre05.g232800) described by Phytozome in Arabidopsis is AT1G31170.1. TargetP does not predict the protein neither to CPL nor to mitochondria.

I guess that the authors mention SRX1 encoded by Cre17.g729950 (Table S1 for example) where shows a clear induction after RB treatment. This particular SRX1 is certainly not predicted to be cytoplasmic and targeted secondary to mitochondria, as suggested by the authors.

The figures and conclusions should be revised accordingly.

Author Response

Comments and Suggestions for Authors

The authors have introduced some modifications which greatly help; however, many imprecisions are still present and should be corrected.

Thanks again for reviewer`s valuable comments and suggestions. We described the parental strain CC4348 with more details, predicted the subcellular localization of APX1 and SRX1, revised the model figure 7 and improved the quality of figure S2.

Point 1: Lines 40-42

‘Photosystem II releases oxygen with H2O, accompanied by the production of ROS, which include singlet oxygen, superoxide, hydrogen peroxide, and hydroxyl radicals.’

Sentence is not clear: PSII does not release both O2 and H2O. In addition, PSI also participates to the production of ROS.

Response 1:

Revised as follows:

Photosystem releases oxygen with H2O, accompanied by the production of ROS, which include singlet oxygen, superoxide, hydrogen peroxide, and hydroxyl radicals.’

Sentence is not clear: PSII does not release both O2 and H2O. In addition, PSI also participates to the production of ROS.

Point 2: Lines 76-77

The description of the sta6 mutant is not made in reference 15. Please use the correct original reference and additional references describing the sta6 mutant. For example, the sta6 mutant has been shown not only defective for the small subunit of the heterotetrameric ADP-glucose pyrophosphorylase, being also affected in one RBO gene.

Please also consider the paper which describes the oxidative stress in the sta6 mutant.

(Sci Rep. 2019 Jul 8;9(1):9856. doi: 10.1038/s41598-019-46313-6. Impairment of starch biosynthesis results in elevated oxidative stress and autophagy activity in Chlamydomonas reinhardtii. Tran QG, Cho K, Park SB, Kim U, Lee YJ, Kim HS.)

The fact that the sta6 mutant is also affected in its stress response should be considered for the comparison with gpx5.

The parental strain CC4348 is not cell wall less in the Chlamy collection. Please check.

Response 2:

The description of the sta6 mutant was made in reference 18-21 and revised as follows:

The parental strain of C. reinhardtii used was CC4348, which was also available as cw15 sta6 (cw15 nit1 NIT2 arg7-7 sta6-1::ARG7 mt+) strain descripted as from the Chlamydomonas Resource Center (http://chlamycollection.org). The sta6 mutant has been shown not only defective for the small subunit of the heterotetrameric ADP-glucose pyrophosphorylase (reference 18, 19), being also affected in respiratory burst oxidase gene (RBO1, Cre03.g188300) (reference 20) and sensitive to oxidative stress levels of lipid peroxidation (reference 21).

  1. Zabawinski, C.; Van Den Koornhuyse, N.; D'Hulst, C., et al., Starchless mutants of Chlamydomonas reinhardtii lack the small subunit of a heterotetrameric ADP-glucose pyrophosphorylase. Journal of bacteriology 2001, 183 (3), 1069-77.
  2. Wang, Z. T.; Ullrich, N.; Joo, S., et al., Algal lipid bodies: stress induction, purification, and biochemical characterization in wild-type and starchless Chlamydomonas reinhardtii. Eukaryotic cell 2009, 8 (12), 1856-68.
  3. Blaby, I. K.; Glaesener, A. G.; Mettler, T., et al., Systems-level analysis of nitrogen starvation-induced modifications of carbon metabolism in a Chlamydomonas reinhardtii starchless mutant. The Plant cell 2013, 25 (11), 4305-23.
  4. Tran, Q. G.; Cho, K.; Park, S. B., et al., Impairment of starch biosynthesis results in elevated oxidative stress and autophagy activity in Chlamydomonas reinhardtii. Scientific reports 2019, 9 (1), 9856.

CC4348 was from Zi Teng Wang, Goodenough lab, Washington University in St Louis, June 2010, which was the same strain as CC-4333 received from the Goodenough lab. This mutant was obtained by insertional mutagenesis of the cw15 arg7 strain CC-4324 (Ball 330). Its parental strain cw15 (330) was originally isolated as a cell wall-less mutant (Davies, D. R., and A. Plaskitt. 1971. Genetical and structural analyses of cell-wall formation in Chlamydomonas reinhardi. Genet. Res. 17:33–43.), so the CC4348 was also a cell wall-less mutant.

Point 3:

Line 289: ‘re-analyzed’? please use ‘analyze’

Response 3:

Revised as follows:

Moreover, we also analyzed the expression change of genes in the chlororespiraton.

Point 4:

Line 460: identity instead of similarity?

Response 4:

Revised as follows:

The identity between 13S-lipoxygenase (13-LOX, Cre12.g512300) in C. reinhardtii to lipoxygenase (LOX1, AT1G55020) in Arabidopsis thaliana, 12-oxophytodienoic acid reductase (OPR, Cre03.g210513) in C. reinhardtii to oxophytodienoate-reductase 3 (OPR3, AT2G06050) in Arabidopsis thaliana is 36% and 46%, respectively

Point 5:

Line 558-562: Blaby et al. 2015 (Plant J, mentioned by the authors ref 42) showed that both PS and respiratory activities are decreased in WT after H2O2 treatment. This suggests that ATP produced by respiration is in less amount in stress conditions. The authors should revise their conclusions (These results suggested that the high concentration of ROS stimulates the mETC to provide more ATP – line 561-562) if they would like to rely on the results of Blaby et al or make respiratory measurements to clarify the point.

Line 561: consumption instead of production? Decreased and not 'decreasd'

Response 5:

Revised as follows:

Transcripts encoding proteins of the most mETC genes not changed and the O2 consumption decreased. This suggested that ATP produced by respiration is in less amount in stress conditions.

Point 6:

As previously mentioned, the main enzyme catalyzing the conversion of P-glycolate into glyoxylate in Chlamydomonas is a glycolate dehydrogenase putatively localized in mitochondria (see Enzymatic characterization of Chlamydomonas reinhardtii glycolate dehydrogenase and its nearest proteobacterial homologue, Mohamed H. Aboelmy, Christoph Peterhansel* Plant Physiology and Biochemistry 79 (2014) 25e30). Please adapt the figures accordingly. What is the FPKM levels of the glycolate dehydrogenase in the different strains and conditions?

Response 6:

The glycolate dehydrogenase (GYD1, Cre06.g288700) catalyzed the conversion of P-glycolate into glyoxylate in Chlamydomonas. The transcript abundance of GYD1 remained no changed from 24 RPKM in CC4348 to 37 RPKM in the gpx5 mutant under laboratory growth conditions.

We also analysed the expression changes of other genes except for GYD1 in the chlororespiraton. Among them, 5 out of 7 transcripts showed no difference in abundance between CC4348 and the gpx5 mutant, including phosphoglycolate phosphatase (PGP1 encoding by Cre03.g168700), serine hydroxymethyltransferase (SHMT1, Cre16.g664550), alanine-glyoxylate transaminase (AGT1 and AGT2) and serine glyoxylate aminotransferase (SGA1). Only hydroxypyruvate reductase (HPR1) and alanine aminotransferase (AAT1) were down-regulated in the gpx5 mutant.

Point 7:

Line 836: ref 42 is not dealing with sak1.

Response 7:

It has been revised in reference 17.

Point 8:

Figure S2: poor quality, remove the background

Response 8:

The quality of figure S2 has been improved as follows. It can also be seen in the supplementary file.

Point 9:

APX1

According to Urzica et al 2012 JBC, APX1 would be located in CPL for removal of H2O2 from PSI in Chlamydomonas. According to Wu and Wang (2019) PLoS ONE 14(12): e0226543. https://doi.org/10.1371/journal. pone.0226543) APX1 would be located to CPL or mitochondria. The localization of APX1 is thus far to be clear. Please discuss and adapt figures according to the possibility that APX1 is not cytosolic.

Response 9:

According to Urzica et al 2012 JBC (Impact of Oxidative Stress on Ascorbate Biosynthesis in Chlamydomonas via Regulation of the VTC2 Gene Encoding a GDP-L-galactose Phosphorylase), APX1 would be located in the chloroplast.

According to Wu and Wang (2019) PLOS ONE (Comparative analysis of ascorbate peroxidases (APXs) from selected plants with a special focus on Oryza sativa employing public databases), APX1 would be located to chloroplast or mitochondria.

Moreover, according to the third reference Pitsch et al., 2010 (Comparison of the chloroplast peroxidase system in the chlorophyte Chlamydomonas reinhardtii, the bryophyte Physcomitrella patens, the lycophyte Selaginella moellendorffii and the seed plant Arabidopsis thaliana), APX1 was located in chloroplast stroma.

We also predicted the location of APX1 in the chloroplast as shown in. revised figure 7.

Point 10:

SRX1

In Phytozome, 2 genes (Cre05.g232800 and Cre17.g729950) are named SRX1. SRX1 encoded by Cre17.g729950 is putatively localized in the CPL. Indeed, the homologue described by Phytozome in Arabidopsis is AT1G31170.1 which has been demonstrated to be chloroplastic (Rey et al Plant J 2007).

In addition, TargetP, mentioned above by the authors, predicts a chloroplast localization for SRX1 (Cre17.g729950) and the paper previously mentioned by the authors (Blably et al 2015 Plant J) also proposed a CPL localization of SRX1.

The homologue for the other SRX1 (Cre05.g232800) described by Phytozome in Arabidopsis is AT1G31170.1. TargetP does not predict the protein neither to CPL nor to mitochondria.

I guess that the authors mention SRX1 encoded by Cre17.g729950 (Table S1 for example) where shows a clear induction after RB treatment. This particular SRX1 is certainly not predicted to be cytoplasmic and targeted secondary to mitochondria, as suggested by the authors.

The figures and conclusions should be revised accordingly.

Response 10:

Yes, SRX1 encoded by Cre17.g729950 showed a clear induction after RB treatment, but in table S3. After treatment with RB for 30 min, the SRX1 encoded by Cre17.g729950 were both up-regulated in CC4348 and the gpx5 mutant. The abundance of SRX1 (Cre17.g729950) were increased from 8 RPKM to 39 RPKM in CC4348, while 13 RPKM to 151 RPKM in the gpx5 mutant. Another SRX1 encoded by Cre05.g232800 remain relative low transcript abundance both in CC4348 and the gpx5 mutant.

According to the reference (Asha Ramesh et al, 2014, Role of sulfiredoxin in systemic diseases influenced by oxidative stress Role of sulfiredoxin in systemic diseases influenced by oxidative stress), sulfiredoxin (SRX1) is primarily located in the cytosol and it gets translocated to the mitochondria during increased oxidative burden, which plays a crucial role in thiol homoeostasis when under oxidative stress.

The figure 7 has been revised.
